# Female-biased upregulation of insulin pathway activity mediates the sex difference in *Drosophila* body size plasticity

Jason W Millington[1], George P Brownrigg[1], Charlotte Chao[1], Ziwei Sun[1], Paige J Basner-Collins[1], Lianna W Wat[1], Bruno Hudry[2†], Irene Miguel-Aliaga[2], Elizabeth J Rideout[1]*

[1]Department of Cellular and Physiological Sciences, Life Sciences Institute, The University of British Columbia, Vancouver, Canada; [2]MRC London Institute of Medical Sciences, and Institute of Clinical Sciences, Faculty of Medicine, Imperial College London, London, United Kingdom

**\*For correspondence:**
elizabeth.rideout@ubc.ca

**Present address:** [†]Institut de Biologie Valrose, Centre de Biochimie, Faculte des Sciences, Universite Nice Sophia Antipolis, Nice, France

**Competing interests:** The authors declare that no competing interests exist.

**Abstract** Nutrient-dependent body size plasticity differs between the sexes in most species, including mammals. Previous work in *Drosophila* showed that body size plasticity was higher in females, yet the mechanisms underlying increased female body size plasticity remain unclear. Here, we discover that a protein-rich diet augments body size in females and not males because of a female-biased increase in activity of the conserved insulin/insulin-like growth factor signaling pathway (IIS). This sex-biased upregulation of IIS activity was triggered by a diet-induced increase in *stunted* mRNA in females, and required *Drosophila insulin-like peptide 2*, illuminating new sex-specific roles for these genes. Importantly, we show that sex determination gene *transformer* promotes the diet-induced increase in *stunted* mRNA via transcriptional coactivator Spargel to regulate the male-female difference in body size plasticity. Together, these findings provide vital insight into conserved mechanisms underlying the sex difference in nutrient-dependent body size plasticity.

## Introduction

In insects, the rate of growth during development is influenced by environmental factors such as nutrient availability (*Boulan et al., 2015*; *Edgar, 2006*; *Hietakangas and Cohen, 2009*; *Nijhout, 2003*; *Nijhout et al., 2014*). When nutrients are abundant, the growth rate is high and body size is large (*Beadle et al., 1938*; *Edgar, 2006*; *Mirth and Shingleton, 2012*; *Nijhout, 2003*; *Robertson, 1963*). When nutrients are scarce, the growth rate is lower and body size is smaller (*Beadle et al., 1938*; *Edgar, 2006*; *Mirth and Riddiford, 2007*; *Mirth and Shingleton, 2012*; *Nijhout, 2003*; *Robertson, 1963*). This ability of an organism or genotype to adjust its body size in line with nutrient availability is a form of phenotypic plasticity (*Agrawal, 2001*; *Garland, 2006*). While the capacity of individuals to display nutrient-dependent changes to body size depends on many factors, one important factor that affects phenotypic plasticity is whether an animal is male or female (*Stillwell et al., 2010*; *Teder and Tammaru, 2005*). For example, in *Drosophila* the magnitude of changes to wing cell size and cell number in a nutrient-poor diet were larger in females compared with males (*Alpatov, 1930*). Similarly, the magnitude of protein- and carbohydrate-induced changes to several morphological traits was larger in female flies (*Shingleton et al., 2017*). While these studies clearly establish a sex difference in nutrient-dependent phenotypic plasticity, the genetic and molecular mechanisms underlying this increased trait size plasticity in females remain unclear.

Clues into potential mechanisms underlying the increased nutrient-dependent phenotypic plasticity in female flies have emerged from over 20 years of studies on nutrient-dependent growth in *Drosophila* (*Andersen et al., 2013*; *Boulan et al., 2015*; *Edgar, 2006*; *Koyama and Mirth, 2018*; *Mirth and Piper, 2017*). In particular, these studies have identified the conserved insulin/insulin-like growth factor signaling pathway (IIS) as a key regulator of nutrient-dependent growth in *Drosophila* (*Böhni et al., 1999*; *Britton et al., 2002*; *Chen et al., 1996*; *Fernandez et al., 1995*; *Grewal, 2009*; *Teleman, 2010*). In nutrient-rich conditions, insulin-producing cells (IPCs) in the larval brain release *Drosophila* insulin-like peptides (Dilps) into the circulation (*Brogiolo et al., 2001*; *Géminard et al., 2009*; *Ikeya et al., 2002*; *Rulifson et al., 2002*). These Dilps bind the Insulin-like Receptor (InR; FBgn0283499) on target cells to induce receptor autophosphorylation and recruitment of adapter proteins (*Almudi et al., 2013*; *Böhni et al., 1999*; *Chen et al., 1996*; *Poltilove et al., 2000*; *Werz et al., 2009*). These adapter proteins enable the recruitment of the regulatory and catalytic subunits of the *Drosophila* homolog of phosphatidylinositol 3-kinase (*Pi3K21B*; FBgn0020622 and *Pi3K92E*; FBgn0015279, respectively), which catalyze the production of phosphatidylinositol (3,4,5)-trisphosphate (PIP$_3$) from phosphatidylinositol (4,5)-bisphosphate (PIP$_2$) (*Leevers et al., 1996*). Increased plasma membrane PIP$_3$ recruits and activates signaling proteins such as phosphoinositide-dependent kinase 1 (Pdk1; FBgn0020386) and Akt (Akt; FBgn0010379), which influence diverse cellular processes to enhance cell, tissue, and organismal size (*Cho et al., 2001*; *Grewal, 2009*; *Rintelen et al., 2001*; *Verdu et al., 1999*).

In contrast, when nutrients are scarce, Dilp release from the IPCs is reduced (*Géminard et al., 2009*), and plasma membrane Pi3K recruitment, PIP$_3$ levels, and Pdk1- and Akt-dependent signaling are all reduced (*Britton et al., 2002*; *Nowak et al., 2013*). Together, these changes diminish cell, tissue, and organismal size (*Arquier et al., 2008*; *Britton et al., 2002*; *Géminard et al., 2009*; *Honegger et al., 2008*; *Okamoto et al., 2013*; *Rulifson et al., 2002*; *Zhang et al., 2009*). Indeed, the potent growth-promoting ability of IIS activation is shown by the fact that increased IIS activity augments body size (*Arquier et al., 2008*; *Goberdhan et al., 1999*; *Honegger et al., 2008*; *Ikeya et al., 2002*; *Nowak et al., 2013*; *Okamoto et al., 2013*; *Oldham et al., 2002*), whereas reduced IIS activity limits cell, organ, and body size (*Böhni et al., 1999*; *Brogiolo et al., 2001*; *Chen et al., 1996*; *Colombani et al., 2003*; *Gao et al., 2000*; *Grönke et al., 2010*; *Leevers et al., 1996*; *Murillo-Maldonado et al., 2011*; *Rulifson et al., 2002*; *Weinkove et al., 1999*; *Zhang et al., 2009*). Because increased IIS activity bypasses the reduction in cell size in low-nutrient conditions (*Britton et al., 2002*; *Géminard et al., 2009*; *Nowak et al., 2013*), and mutations that blunt IIS pathway activity reduce size in nutrient-rich contexts (*Böhni et al., 1999*; *Brogiolo et al., 2001*; *Chen et al., 1996*; *Leevers et al., 1996*), *Drosophila* studies have established IIS as one key pathway that promotes organismal growth downstream of nutrient input. While this highlights the impact of *Drosophila* on our knowledge of how IIS couples nutrient input with growth, it is important to note that most studies used a mixed-sex population of larvae. Given that cell and body size differ significantly between male and female flies (*Alpatov, 1930*; *Brown and King, 1961*; *Okamoto et al., 2013*; *Partridge et al., 1994*; *Rideout et al., 2015*; *Sawala and Gould, 2017*; *Testa et al., 2013*), more knowledge is needed of nutrient-dependent changes to body size and IIS activity in each sex.

Recent studies have begun to make progress in this area by studying IIS regulation and function in both sexes in a single dietary context (reviewed in *Millington and Rideout, 2018*). For example, in late third instar larvae, there are sex differences in *dilp* mRNA levels, IIS activity, and *Drosophila* insulin-like peptide 2 (Dilp2; FBgn0036046) secretion from the IPCs (*Rideout et al., 2015*; *McDonald et al., 2020*). Similarly, transcriptomic studies have detected male-female differences in mRNA levels of genes associated with IIS function (*Mathews et al., 2017*; *Rideout et al., 2015*), and revealed links between IIS and the sex determination hierarchy gene regulatory network (*Castellanos et al., 2013*; *Chang et al., 2011*; *Clough et al., 2014*; *Fear et al., 2015*; *Garner et al., 2018*; *Goldman and Arbeitman, 2007*). As increasing evidence of sex-specific IIS regulation accumulates, several reports reveal sex-limited and sex-biased phenotypic effects caused by changes to IIS function. Changes to IIS activity in larvae show sex-biased effects on growth and final body size (*Grönke et al., 2010*; *Rideout et al., 2015*; *Shingleton et al., 2005*; *Testa et al., 2013*; *Millington et al., 2021*), and there are widespread sex-specific and sex-biased changes to gene expression in adult flies with altered diet and IIS activity (*Camus et al., 2019*; *Graze et al., 2018*). Further, sex differences exist in how changes to diet and IIS activity affect life span (*Bjedov et al., 2010*; *Clancy, 2001*; *Giannakou et al., 2004*; *Grönke et al., 2010*; *Regan et al., 2016*; *Tatar et al.,*

*2001*; *Woodling et al., 2020*; *Wu et al., 2020*). Together, these studies illuminate the utility of *Drosophila* in revealing sex-specific IIS regulation and describing the physiological impact of this regulation. Yet, more studies are needed to discover the molecular mechanisms underlying sex-specific IIS regulation, and to extend these studies beyond a single nutritional context.

Additional insights into male-female differences in the regulation of cell, tissue, and body size arise from studies on sex determination genes. In *Drosophila*, sex is determined by the number of X chromosomes. In XX females, a functional splicing factor called Sex-lethal (Sxl; FBgn0264270) is produced (*Bell et al., 1988*; *Bridges, 1921*; *Cline, 1978*; *Salz and Erickson, 2010*). Sxl-dependent splicing of *transformer* (*tra*; FBgn0003741) pre-mRNA allows a functional Tra protein to be produced in females (*Belote et al., 1989*; *Boggs et al., 1987*; *Inoue et al., 1990*; *Sosnowski et al., 1989*). In XY males, the lack of a functional Sxl protein causes the default splicing of *tra* pre-mRNA, and no functional Tra protein is produced in males (*Cline and Meyer, 1996*; *Salz and Erickson, 2010*; *Belote et al., 1989*; *Boggs et al., 1987*; *Inoue et al., 1990*; *Sosnowski et al., 1989*). The presence of functional Sxl and Tra proteins in females account for most aspects of female sexual development, behavior, and physiology (*Anand et al., 2001*; *Billeter et al., 2006*; *Brown and King, 1961*; *Camara et al., 2008*; *Christiansen et al., 2002*; *Clough et al., 2014*; *Dauwalder, 2011*; *Demir and Dickson, 2005*; *Goodwin et al., 2000*; *Hoshijima et al., 1991*; *Hudry et al., 2016*; *Hudry et al., 2019*; *Ito et al., 1996*; *Millington and Rideout, 2018*; *Neville et al., 2014*; *Nojima et al., 2014*; *Pavlou et al., 2016*; *Pomatto et al., 2017*; *Regan et al., 2016*; *Rezával et al., 2014*; *Rezával et al., 2016*; *Rideout et al., 2010*; *Ryner et al., 1996*; *Sturtevant, 1945*; *von Philipsborn et al., 2014*). Recently, new roles for Sxl and Tra in regulating body size were also described. While female flies are normally larger than males, females lacking neuronal *Sxl* were smaller than control females, and not different in size from males (*Sawala and Gould, 2017*). Similarly, females lacking a functional Tra protein were smaller than control females; however, these *tra* mutant females were still larger than males (*Brown and King, 1961*; *Mathews et al., 2017*; *Rideout et al., 2015*). Together, these studies indicate that Tra and Sxl are required to promote a larger body size in females; however, much remains to be discovered about the mechanisms by which Sxl and Tra impact body size. Moreover, which sex determination genes contribute to the male-female difference in diet-induced trait size plasticity remains unknown, as studies on sex determination genes used a single diet.

In the present study, we aimed to improve knowledge of the genetic and molecular mechanisms that contribute to male-female differences in nutrient-dependent phenotypic plasticity in *Drosophila*. Our detailed examination of body size revealed increased phenotypic plasticity in females in response to a protein-rich diet, in line with studies on plasticity in other traits (*Shingleton et al., 2017*). We discovered that a female-biased upregulation of IIS activity was responsible for the larger body size of females raised on a protein-rich diet. Mechanistically, we show that the nutrient-dependent upregulation of *stunted* (*sun*; FBgn0014391) mRNA levels by transcriptional coactivator Spargel (Srl; FBgn0037248) in females triggers the diet-induced increase in IIS activity, as females with reduced *sun* do not augment IIS activity or body size in a protein-rich diet. Importantly, we show that sex determination gene *tra* is required for the nutrient-dependent increase in *sun* mRNA, IIS activity, and phenotypic plasticity in females, and that Srl represents a key link between Tra and regulation of *sun* mRNA levels. In males, ectopic Tra expression confers nutrient-dependent body size plasticity via Srl-mediated regulation of *sun* mRNA levels and IIS activity. Together, these results provide new insight into the molecular mechanisms that govern male-female differences in body size plasticity, and identify a previously unrecognized role for sex determination gene *tra* in regulating nutrient-dependent phenotypic plasticity.

## Results

### High levels of dietary protein are required for increased nutrient-dependent body size plasticity in females

Previous studies identified a sex difference in nutrient-dependent plasticity in several morphological traits (*Shingleton et al., 2017*; *Stillwell et al., 2010*; *Teder and Tammaru, 2005*). To determine whether sex differences in nutrient-dependent body size plasticity exist in *Drosophila*, we measured pupal volume, an established readout for *Drosophila* body size (*Delanoue et al., 2010*), in *white*[1118] (*w*; FBgn0003996) males and females reared on diets of varying nutrient quantity. We found that

pupal volume in $w^{1118}$ female larvae raised on the two-acid diet (1X) (*Lewis, 1960*) was significantly larger than genotype-matched females raised on a diet with half the nutrient quantity (0.5X) (*Figure 1—figure supplement 1A*). In $w^{1118}$ males, pupal volume was also significantly larger in larvae raised on the 1X diet compared with the 0.5X diet (*Figure 1—figure supplement 1A*). No significant sex-by-diet interaction was detected using a two-way analysis of variance (ANOVA) (sex:diet interaction p=0.7048; *Supplementary file 1*), suggesting that nutrient-dependent body size plasticity was not different between the sexes in this context. We next compared pupal volume in $w^{1118}$ males and females raised on the 1X diet with larvae cultured on a diet with twice the nutrient content (2X). Pupal volume in $w^{1118}$ females was significantly larger in larvae raised on the 2X diet compared with larvae cultured on the 1X diet (*Figure 1—figure supplement 1A*). In $w^{1118}$ males, the magnitude of the nutrient-dependent increase in pupal volume was smaller compared with female larvae (*Figure 1—figure supplement 1A*; sex:diet interaction p<0.0001; *Supplementary file 1*). This suggests that in nutrient-rich conditions, there is a sex difference in phenotypic plasticity, where nutrient-dependent body size plasticity is higher in females. To represent the normal body size responses of each sex to nutrient quantity, we plotted reaction norms for pupal volume in $w^{1118}$ males and females raised on different diets (*Figure 1—figure supplement 1B*). The body size response to increased nutrient quantity between 0.5X and 1X was not different between the sexes (*Figure 1—figure supplement 1B*); however, the body size response to increased nutrient quantity between 1X and 2X was larger in females than in males (*Figure 1—figure supplement 1B*). Importantly, these findings were not specific to pupal volume, as we reproduced our findings using adult weight as an additional readout for body size (*Figure 1A,B*). Thus, our findings demonstrate that while phenotypic plasticity is similar between the sexes in some nutritional contexts, body size plasticity is higher in females than in males in a nutrient-rich environment.

To narrow down macronutrients that account for the increased body size plasticity in females, we changed individual food ingredients and measured body size in $w^{1118}$ males and females. We first altered dietary yeast, as previous studies show that yeast is a key source of protein and an important determinant of larval growth (*Britton et al., 2002*; *Géminard et al., 2009*; *Robertson, 1963*). In $w^{1118}$ females raised on a diet with yeast content that corresponds to the amount in the 2X diet (2Y diet), pupal volume was significantly larger than in females raised on a diet containing half the yeast content (1Y) (*Figure 1—figure supplement 1C*). It is important to note that the yeast and calorie content of the 1Y diet was within the range of standard diets used in many larval growth studies (22.65 g/L vs. 21–46 g/L and 586 calories/L vs 459–760 calories/L, respectively) (*Ghosh et al., 2014*; *Koyama and Mirth, 2016*; *Marshall et al., 2012*; *Sawala and Gould, 2017*), and therefore does not represent a nutrient-restricted diet. In $w^{1118}$ males, the magnitude of the nutrient-dependent increase in pupal volume was smaller than in females (*Figure 1—figure supplement 1C*; sex:diet interaction p=0.0001; *Supplementary file 1*), suggesting that nutrient-dependent body size plasticity was higher in females in a yeast-rich context. Indeed, when we plotted reaction norms for pupal volume in both sexes, the magnitude of the yeast-dependent change in pupal volume (*Figure 1—figure supplement 1D*) and adult weight (*Figure 1C,D*) was larger in females than in males. This sex difference in phenotypic plasticity in a yeast-rich context was reproduced in *Canton-S* (*CS*), a wild-type strain (*Figure 1—figure supplement 2A,B*), and using wing length as an additional measure of size (*Figure 1—figure supplement 3A*). Thus, our findings indicate that the male-female difference in nutrient-dependent body size plasticity persists across multiple genetic backgrounds, and confirms that body size is a robust trait to monitor nutrient-dependent phenotypic plasticity.

Given the sex difference in body size plasticity in response to altered yeast content, we hypothesized that yeast may trigger increased nutrient-dependent body size plasticity in females. To test this, we raised larvae on diets with altered sugar (*Figure 1—figure supplement 4A*) or calorie content (*Figure 1—figure supplement 4B*). Because we observed no sex:diet interaction for either manipulation (sex:diet interaction p=0.6536 and p=0.3698, respectively; *Supplementary file 1*), this suggests dietary yeast mediates the sex difference in nutrient-dependent body size plasticity. To test whether protein is the macronutrient in yeast that enables sex-specific phenotypic plasticity, we pharmacologically limited protein breakdown by culturing larvae on the 2Y diet supplemented with either a broad-spectrum protease inhibitor (protease inhibitor cocktail; PIC) or a serine protease-specific inhibitor (4-(2-aminoethyl)benzenesulfonyl fluoride hydrochloride; AEBSF). Previous studies suggest that these inhibitors are specific, as the growth-inhibitory effect of these protease inhibitors was buffered by feeding larvae with bacteria that enhance intestinal protease mRNA levels and gut

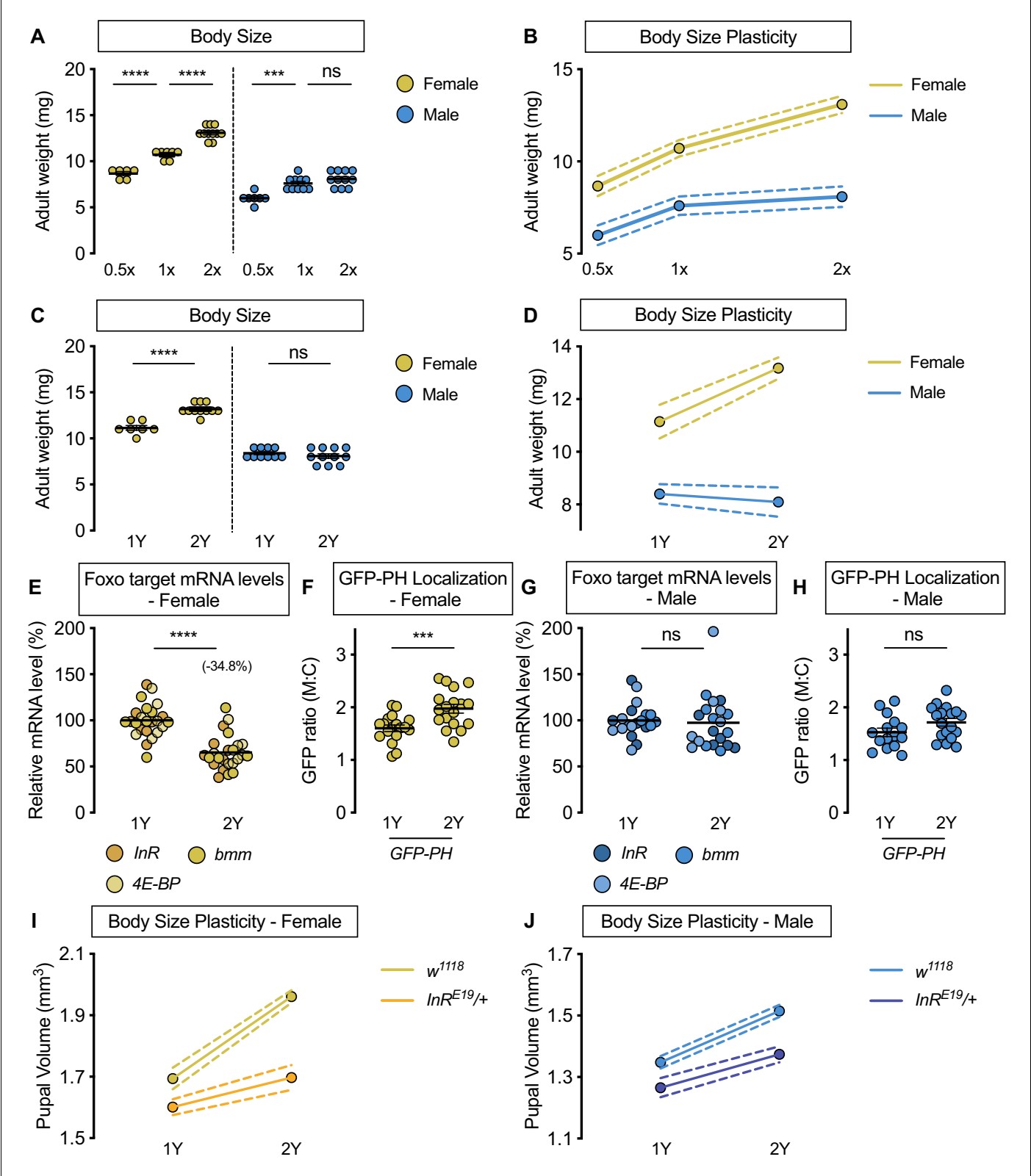

**Figure 1.** Upregulation of IIS activity is required for increased nutrient-dependent body size plasticity in females in a protein-rich diet. (A) Adult weight was significantly higher in $w^{1118}$ males and females cultured on 1X compared with flies raised on 0.5X (p<0.0001 for both sexes; two-way ANOVA followed by Tukey HSD test). The magnitude of this increase in adult weight was the same in both sexes (sex:diet interaction p=0.3197; two-way ANOVA followed by Tukey HSD test). Adult weight was significantly higher in $w^{1118}$ females raised on 2X compared to flies cultured on 1X; however,

*Figure 1 continued on next page*

*Figure 1 continued*

male adult weight was not significantly increased (p<0.0001 and p=0.4015, respectively; two-way ANOVA followed by Tukey HSD test), where the diet-dependent increase in adult weight was higher in females (sex:diet interaction p=0.0003; two-way ANOVA followed by Tukey HSD test). (B) Reaction norms for adult weight in response to changes in nutrient quantity in $w^{1118}$ females and males, plotted using the data presented in panel A. n = 6–11 groups of 10 flies. (C) Adult weight was significantly higher in females cultured on 2Y compared with flies raised on 1Y; however, male adult weight was not significantly higher in flies raised on 2Y compared with males cultured on 1Y (p<0.0001 and p=0.7199, respectively; two-way ANOVA followed by Tukey HSD test, sex:diet interaction p<0.0001). (D) Reaction norms for adult weight in $w^{1118}$ females and males reared on either 1Y or 2Y, plotted using data from panel C. n = 7–11 groups of 10 flies. (E) In females, mRNA levels of Foxo targets (*insulin receptor (InR), brummer (bmm)*, and *eukaryotic initiation factor 4E-binding protein (4E-BP)*), were significantly lower in larvae raised on a protein-rich diet (2Y) compared with larvae raised on a diet containing half the protein content (1Y) (p<0.0001; Student's *t* test). n = 8 biological replicates. (F) Quantification of the ratio between cell surface membrane-associated green fluorescent protein (GFP) and cytoplasmic GFP (GFP ratio [M:C]) in a dissected fat body of female larvae from the GFP-PH strain. The ratio was significantly higher in female larvae cultured on 2Y compared with larvae raised on 1Y (p=0.001; Student's *t* test). n = 18 biological replicates. (G) In males, there was no significant difference in mRNA levels of Foxo targets between larvae raised on 2Y compared with larvae cultured on 1Y (p=0.7323; Student's *t* test). n = 6–7 biological replicates. (H) In males, the M:C ratio for GFP-PH was not significantly different between males cultured on 2Y compared with larvae raised on 1Y (p=0.0892; Student's *t* test). n = 15–18 biological replicates. (I) Pupal volume was significantly higher in both $w^{1118}$ females and $InR^{E19}/+$ females reared on 2Y compared with genotype-matched females cultured on 1Y (p<0.0001 for both genotypes; two-way ANOVA followed by Tukey HSD test); however, the magnitude of the nutrient-dependent increase in pupal volume was lower in $InR^{E19}/+$ females (genotype:diet interaction p<0.0001; two-way ANOVA followed by Tukey HSD test). n = 58–77 pupae. (J) Pupal volume was significantly higher in both $w^{1118}$ males and $InR^{E19}/+$ males reared on 2Y compared with genotype-matched males cultured on 1Y (p<0.0001 for both genotypes; two-way ANOVA followed by Tukey HSD test). While we observed a sex:diet interaction in the $w^{1118}$ control genotype, there was no sex:diet interaction in the $InR^{E19}/+$ genotype (p<0.0001 and p=0.7104, respectively; two-way ANOVA followed by Tukey HSD test). n = 47–76 pupae. For body size plasticity graphs, filled circles indicate mean body size, and dashed lines indicate 95% confidence interval. *** indicates p<0.001, **** indicates p<0.0001; ns indicates not significant; error bars indicate SEM.

The online version of this article includes the following figure supplement(s) for figure 1:

**Figure supplement 1.** Increased female body size plasticity in a protein-rich diet.
**Figure supplement 2.** Increased nutrient-dependent body size plasticity in *Canton-S* females.
**Figure supplement 3.** Increased nutrient-dependent plasticity in female wing size.
**Figure supplement 4.** No sex-specific effect of altering dietary sugar concentration or calorie content.
**Figure supplement 5.** Pharmacological inhibition of protein breakdown has female-biased effects on body size.
**Figure supplement 6.** No sex difference in food intake or time to pupation.
**Figure supplement 7.** Larger body size does not confer increased body size plasticity.

proteolytic activity (*Erkosar et al., 2015*). While we found a significant body size reduction in both sexes treated with protease inhibitors (*Figure 1—figure supplement 5A,B*), in line with previous studies (*Erkosar et al., 2015*), the magnitude of the inhibitor-induced decrease in pupal volume was larger in female larvae than in males (sex:treatment interaction p=0.0029 [PIC] and p<0.0001 [AEBSF]; *Supplementary file 1*). This indicates that yeast-derived dietary protein is the macronutrient that augments nutrient-dependent body size plasticity in females. While two potential explanations for the male-female difference in body size plasticity are a sex difference in food intake or length of the growth period, we found no differences in either phenotype between $w^{1118}$ male and female larvae cultured on 1Y or 2Y (*Figure 1—figure supplement 6A–C*). Moreover, the larger body size of female larvae does not explain their increased nutrient-dependent body size plasticity, as a genetic manipulation that augments male body size did not enhance phenotypic plasticity (*Figure 1—figure supplement 7A,B*). Taken together, our data reveals female larvae have enhanced body size plasticity in a nutrient-rich context, and identifies abundant dietary protein as a prerequisite for females to maximize body size.

## The nutrient-dependent upregulation of IIS activity in females is required to achieve a larger body size in a protein-rich context

In a mixed-sex population of *Drosophila* larvae, IIS activity is positively regulated by nutrient availability to promote growth (*Böhni et al., 1999*; *Britton et al., 2002*; *Chen et al., 1996*; *Fernandez et al., 1995*; *Grewal, 2009*; *Teleman, 2010*). We therefore examined nutrient-dependent changes to IIS activity in larvae raised on 1Y and 2Y (*Figure 1E–H*). Previous studies show that high levels of IIS activity repress mRNA levels of several genes via transcription factor Forkhead box, sub-group O (Foxo; FBgn0038197) (*Alic et al., 2011*; *Jünger et al., 2003*; *Kang et al., 2017*; *Puig and Tjian, 2005*; *Zinke et al., 2002*). We therefore assessed mRNA levels of known Foxo target

genes *InR*, *brummer* (*bmm*, FBgn0036449), and *eukaryotic initiation factor 4E-binding protein* (*4E-BP*, FBgn0261560) together to quantify IIS activity in each sex and dietary context, an established approach to analyze coregulated genes (*Blaschke et al., 2013*; *Hudry et al., 2019*). In $w^{1118}$ females, mRNA levels of Foxo target genes were significantly lower in larvae reared on 2Y than in larvae raised on 1Y (*Figure 1E*). This suggests IIS activity is significantly higher in females raised on 2Y than in females cultured on 1Y. To confirm this, we used the localization of a ubiquitously-expressed green fluorescent protein (GFP) fused to a pleckstrin homology (PH) domain (GFP-PH) as an additional readout of IIS activity. Because high levels of IIS activity raise plasma membrane $PIP_3$, and PH domains bind specifically to $PIP_3$, larvae with elevated IIS activity show increased membrane localization of GFP-PH (*Britton et al., 2002*). We observed a significantly higher membrane localization of GFP-PH in females cultured on 2Y than in female larvae raised on 1Y (*Figure 1F*). Together with increased Foxo target gene repression in 2Y, this GFP-PH data indicates that females reared on 2Y have higher IIS activity than females cultured on 1Y. In males, the magnitude of the nutrient-dependent change in Foxo target genes was smaller than in females (*Figure 1G*), as we detected a significant sex:diet interaction for Foxo target genes (p=0.0007; *Supplementary file 1*). Indeed, there was no significant increase in GFP-PH membrane localization between males raised on 2Y and males reared on 1Y (*Figure 1H*). Taken together, these results reveal a previously unrecognized female-biased upregulation of IIS activity in a protein-rich context.

To determine whether increased IIS activity is required in females for the ability to maximize body size on a protein-rich diet, we measured pupal volume in larvae heterozygous for a hypomorphic mutation in the *InR* gene ($InR^{E19}$/+) that were raised in either 1Y or 2Y. Previous studies have shown that while overall growth is largely normal in $InR^{E19}$/+ heterozygous animals, growth that requires high levels of IIS activity is blunted (*Chen et al., 1996*; *Rideout et al., 2012*; *Rideout et al., 2015*). In $w^{1118}$ control females, larvae cultured on 2Y were significantly larger than larvae raised on 1Y (*Figure 1I*); however, the magnitude of this protein-dependent increase in pupal volume was smaller in $InR^{E19}$/+ females (*Figure 1I*; genotype:diet interaction p<0.0001; *Supplementary file 1*). This suggests that nutrient-dependent body size plasticity was reduced in $InR^{E19}$/+ females. Indeed, while we observed a sex difference in phenotypic plasticity in the $w^{1118}$ control genotype (sex:diet interaction *p*<0.0001 *Supplementary file 1*), the sex difference in nutrient-dependent body size plasticity was abolished in the $InR^{E19}$/+ genotype (*Figure 1I,J*; sex:diet interaction p=0.7104; *Supplementary file 1*). Together, these results indicate that the nutrient-dependent upregulation of IIS activity in females is required for them to achieve a larger body size in a protein-rich context, and that the sex difference in body size plasticity arises from the female-biased upregulation of IIS activity in a protein-rich context.

*dilp2* is required for the nutrient-dependent upregulation of IIS activity and a larger body size in females raised on a protein-rich diet.

Previous studies have identified changes to the production and release of Dilps as important mechanisms underlying nutrient-dependent changes to IIS activity and body size (*Colombani et al., 2003*; *Géminard et al., 2009*; *Zhang et al., 2009*). For example, mRNA levels of *Drosophila insulin-like peptide 3* (*dilp3*; FBgn0044050) and *Drosophila insulin-like peptide 5* (*dilp5*; FBgn0044048), but not *dilp2*, decrease in response to nutrient withdrawal (*Colombani et al., 2003*; *Géminard et al., 2009*; *Ikeya et al., 2002*), and the release of Dilps 2, 3, and 5 from the IPCs is altered by changes in nutrient availability (*Géminard et al., 2009*; *Kim and Neufeld, 2015*). Levels of Dilp2 also fluctuate during larval development (*Slaidina et al., 2009*). Interestingly, a recent study suggests that late third-instar female larvae have increased Dilp2 secretion compared with age-matched males when the larvae were raised in a diet equivalent to 2Y (*Rideout et al., 2015*). Given that Dilp2 is an important growth-promoting Dilp (*Grönke et al., 2010*; *Ikeya et al., 2002*), we tested whether *dilp2* was required in females for the nutrient-dependent upregulation of IIS activity. In control $w^{1118}$ females, mRNA levels of Foxo target genes were significantly lower in larvae raised on 2Y than in larvae reared on 1Y (*Figure 2A*), suggesting a nutrient-dependent increase in IIS activity. In contrast, mRNA levels of Foxo target genes were not significantly lower in *dilp2* mutant female larvae raised on 2Y compared with genotype-matched females cultured on 1Y (*Figure 2A*), suggesting that loss of *dilp2* in females eliminated the nutrient-dependent increase in IIS activity. The magnitude of the nutrient-dependent decrease in Foxo target gene expression was smaller in $w^{1118}$ males compared with $w^{1118}$ females (*Figure 2B*, sex:diet interaction p=0.0511; *Supplementary file 1*), but not in *dilp2* mutant males compared with genotype-matched females (sex:diet interaction p=0.6754;

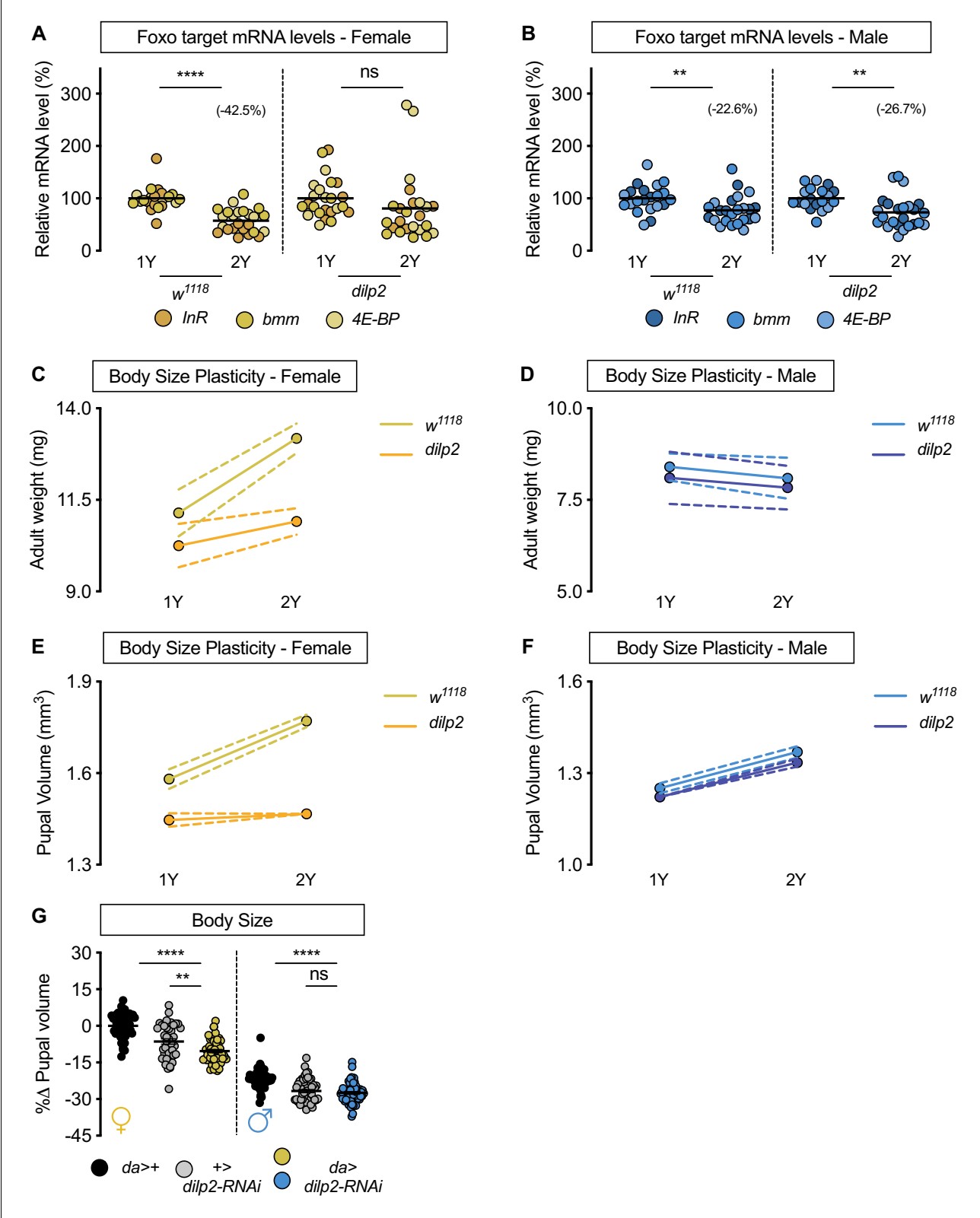

**Figure 2.** *Drosophila* insulin-like peptide 2 is required for the nutrient-dependent upregulation of insulin pathway activity and increased female body size plasticity. (A) In control *w^1118* females, mRNA levels of Foxo targets (*insulin receptor* (*InR*), *brummer* (*bmm*), and *eukaryotic initiation factor 4E-binding protein* (*4E-BP*)), were significantly lower in larvae cultured on a protein-rich diet (2Y) compared with larvae raised on a diet containing half the protein (1Y) (p<0.0001; Student's *t* test). In *dilp2* mutant females, there was no significant difference in mRNA levels of Foxo targets in larvae cultured

*Figure 2 continued on next page*

Figure 2 continued

on 2Y compared with larvae raised on 1Y (p=0.2231 Student's *t* test). n = 8 biological replicates. (B) In control $w^{1118}$ and *dilp2* mutant males, mRNA levels of Foxo targets were significantly lower in larvae cultured on 2Y compared with larvae raised on 1Y (p=0.0066 and p=0.0023 respectively; Student's *t* test). n = 7–8 biological replicates; however, the magnitude of the reduction in Foxo target gene expression in $w^{1118}$ males was smaller than in genotype-matched females. (C) Adult weight was significantly higher in $w^{1118}$ females raised on 2Y compared with flies cultured on 1Y (p<0.0001; two-way ANOVA followed by Tukey HSD test); however, adult weight was not significantly different between *dilp2* mutant females reared on 2Y versus 1Y (p=0.1263; two-way ANOVA followed by Tukey HSD test). n = 7–11 groups of 10 flies. (D) Adult weight in control $w^{1118}$ and *dilp2* mutant males was not significantly higher in flies reared on 2Y compared with males raised on 1Y (p=0.8366 and p=0.8817, respectively; two-way ANOVA followed by Tukey HSD test). There was a significant sex:diet interaction in the control $w^{1118}$ genotype (p<0.0001), but not in the *dilp2* mutant genotype (p=0.0827; two-way ANOVA followed by Tukey HSD test). n = 10–12 groups of 10 flies. (E) Pupal volume was significantly higher in $w^{1118}$ females but not in *dilp2* mutant females reared on 2Y compared with genotype-matched females cultured on 1Y (p<0.0001 and p=0.6486 respectively; two-way ANOVA followed by Tukey HSD test). The magnitude of the nutrient-dependent increase in pupal volume was higher in $w^{1118}$ females (genotype:diet interaction p<0.0001; two-way ANOVA followed by Tukey HSD test). n = 74–171 pupae. (F) Pupal volume was significantly higher in $w^{1118}$ males and *dilp2* mutant males reared on 2Y compared with genotype-matched males cultured on 1Y (p<0.0001 for both genotypes; two-way ANOVA followed by Tukey HSD test). The magnitude of the nutrient-dependent increase in pupal volume was not different between genotypes (genotype:diet interaction p=0.6891; two-way ANOVA followed by Tukey HSD test). n = 110–135 pupae. (G) Pupal volume was significantly reduced in females upon RNAi-mediated knockdown of *dilp2* in 2Y when compared to both control genotypes (p<0.0001 [*da>+*], and p=0.002 [*+>UAS-dilp2-RNAi*], respectively; two-way ANOVA followed by Tukey HSD test), but not in males in 2Y (p<0.0001 [*da>+*], and 0.9634 [*+>UAS-dilp2-RNAi*], respectively; two-way ANOVA followed by Tukey HSD test). The magnitude of the effect of RNAi-mediated knockdown of *dilp2* on pupal volume was higher in females (sex:genotype interaction p=0.003; two-way ANOVA followed by Tukey HSD test). n = 44–59 pupae. For all body size plasticity graphs, filled circles indicate mean body size, and dashed lines indicate 95% confidence interval. ** indicates p<0.01, **** indicates p<0.0001; ns indicates not significant; error bars indicate SEM.

The online version of this article includes the following figure supplement(s) for figure 2:

**Figure supplement 1.** No sex difference in food intake in *dilp2* mutant larvae.

**Figure supplement 2.** HA- and FLAG-tagged *dilp2* transgenic flies exhibit impaired nutrient-dependent body size plasticity.

**Figure supplement 3.** Genotype-dependent changes to *dilp* mRNA levels.

**Figure supplement 4.** Diet-dependent changes to *dilp* mRNA levels.

*Supplementary file 1*). This indicates that *dilp2* loss blocks the female-biased upregulation of IIS activity in a protein-rich diet.

To determine whether the inability to augment IIS activity on 2Y affects the nutrient-dependent increase in female body size, we measured body size in $w^{1118}$ and *dilp2* mutant larvae cultured on either 1Y or 2Y. In $w^{1118}$ control females, adult weight was significantly higher in flies cultured on 2Y compared with flies raised on 1Y (*Figure 2C*); however, this nutrient-dependent increase in adult weight was not observed in *dilp2* mutant females (*Figure 2C*; genotype:diet interaction p=0.0024; *Supplementary file 1*). In $w^{1118}$ control males and *dilp2* mutant males, there was no significant increase in adult weight in flies raised on 2Y compared with genotype-matched flies cultured on 1Y (*Figure 2D*; genotype:diet interaction p=0.935; *Supplementary file 1*). Indeed, in contrast to the sex difference in nutrient-dependent body size plasticity in the $w^{1118}$ genotype (sex:diet interaction p<0.0001; *Supplementary file 1*), the sex difference in phenotypic plasticity was abolished in the *dilp2* mutant genotype (sex:diet interaction p=0.0827; *Supplementary file 1*). Importantly, we replicated all these findings using pupal volume (*Figure 2E,F*), reproduced the female-specific effects of *dilp2* loss by globally overexpressing a *UAS-dilp2-RNAi* transgene (*Figure 2G*), and show that *dilp2* loss does not alter feeding behavior (*Figure 2—figure supplement 1A*). While we did not determine a sex difference in circulating Dilp2 levels in larvae with an endogenously tagged *dilp2* allele due to body size plasticity defects in this strain (*Park et al., 2014*; *Figure 2—figure supplement 2A,B*), an experiment that will be important to repeat in future using alternative ways of measuring circulating Dilp2, we show that changes to *dilp* mRNA levels in males and females lacking *dilp2* (*Figure 2—figure supplement 3A,B*), and nutrient-dependent changes to *dilp* mRNA levels (*Figure 2—figure supplement 4A,B*), were similar in both sexes. Together, our data reveals a previously unrecognized female-specific requirement for *dilp2* in triggering a nutrient-dependent increase in IIS activity and body size in a protein-rich context.

## A nutrient-dependent increase in *stunted* mRNA levels is required for enhanced IIS activity and a larger body size in females cultured in a protein-rich diet

Nutrient-dependent changes in Dilp secretion from the IPCs, and consequently IIS activity, are mediated by humoral factors that are regulated by dietary nutrients (*Britton and Edgar, 1998*; *Delanoue et al., 2016*; *Koyama and Mirth, 2016*; *Rajan and Perrimon, 2012*; *Rodenfels et al., 2014*; *Sano et al., 2015*). For example, in a mixed-sex population of larvae, dietary protein augments mRNA levels of *Growth-blocking peptides 1* and *2* (*Gbp1*, FBgn0034199; *Gbp2*, FBgn0034200), *CCHamide-2* (*CCHa2*; FBgn0038147), *unpaired 2* (*upd2*; FBgn0030904), and *sun* (*Delanoue et al., 2016*; *Koyama and Mirth, 2016*; *Rajan and Perrimon, 2012*; *Sano et al., 2015*). Increased levels of these humoral factors promote the secretion of IPC-produced Dilps to enhance IIS activity and growth (*Delanoue et al., 2016*; *Koyama and Mirth, 2016*; *Meschi et al., 2019*; *Rajan and Perrimon, 2012*; *Sano et al., 2015*). To determine whether any humoral factors contribute to the sex-biased increase in IIS activity in a protein-rich diet, we examined mRNA levels of each factor in larvae of both sexes raised on either 1Y or 2Y. In $w^{1118}$ females, *sun* mRNA levels in larvae reared on 2Y were significantly higher than in larvae cultured on 1Y (*Figure 3A*). In contrast, mRNA levels of *Gbp1*, *Gbp2*, *CCHa2*, and *upd2* were not significantly higher in female larvae reared on 2Y compared with 1Y (*Figure 3B*). Thus, while previous studies have shown that mRNA levels of all humoral factors were severely reduced by a nutrient-restricted diet or nutrient withdrawal (*Delanoue et al., 2016*; *Koyama and Mirth, 2016*; *Rajan and Perrimon, 2012*; *Sano et al., 2015*), our study suggests that for most factors, augmenting dietary protein beyond a widely used level does not further enhance mRNA levels. In males, there was no significant increase in *sun* mRNA levels (*Figure 3C*), or any other humoral factors (*Figure 3D*), in larvae reared on 2Y compared with 1Y. Thus, there is a previously unrecognized sex difference in the regulation of *sun* mRNA levels in a protein-rich context, which we confirm leads to a sex difference in circulating Sun levels (*Figure 3—figure supplement 1A*).

Given that a comprehensive series of genetic, molecular, and organ co-culture experiments have established that Sun promotes IIS activity by enhancing Dilp2 secretion (*Delanoue et al., 2016*), we hypothesized that the female-specific increase in *sun* mRNA levels in 2Y triggers the nutrient-dependent upregulation of IIS activity in females. To test this, we overexpressed UAS-*sun-RNAi* in the larval fat body using *r4-GAL4*, and cultured the animals on either 1Y or 2Y. Importantly, overexpression of the *UAS-sun-RNAi* transgene significantly decreased *sun* mRNA levels in both sexes (*Figure 3—figure supplement 2A,B*), where GAL4 expression was similar between the sexes in 1Y and 2Y (*Figure 3—figure supplement 2C*). In control *r4>+* and *+>UAS-sun-RNAi* females, we observed a significant decrease in Foxo target gene expression in larvae cultured on 2Y compared with genotype-matched larvae reared on 1Y (*Figure 3E*). In contrast, the nutrient-dependent decrease in Foxo target gene expression was absent in *r4>UAS-sun-RNAi* females (*Figure 3E*; diet:genotype interaction p<0.0001; *Supplementary file 1*), suggesting *sun* is required in females for the nutrient-dependent increase in IIS activity. In males, the magnitude of the nutrient-dependent decrease in Foxo target gene expression was smaller than in genotype-matched females for the *r4>+* and *+>UAS-sun-RNAi* control strains (p=0.0166 [*r4>+*]; p=0.0119 [*+>UAS-sun-RNAi*]; *Supplementary file 1*), but not in the *r4>UAS-sun-RNAi* strain (*Figure 3F*) (sex:diet interaction p=0.1121 [*r4>UAS-sun-RNAi*]; *Supplementary file 1*). Importantly, the lack of a diet:genotype interaction among males indicates that there was no effect of genotype on Foxo target gene expression (p=0.1068; *Supplementary file 1*). Together, this data suggests that in females a protein-rich diet stimulates a nutrient-dependent increase in *sun* mRNA that promotes IIS activity. In males, the 2Y diet did not augment *sun* mRNA levels, suggesting one reason for the female-biased increase in IIS activity in a protein-rich diet.

We next asked whether the female-specific increase in *sun* mRNA and its impact on IIS activity contribute to the nutrient-dependent increase in female body size in a protein-rich context. In *r4>+* and *+>UAS-sun-RNAi* control females, adult weight was significantly higher in flies cultured on 2Y compared with genotype-matched flies raised on 1Y (*Figure 3G*). In contrast, the nutrient-dependent increase in adult weight was abolished in *r4>UAS-sun-RNAi* females (*Figure 3G*; genotype:diet interaction p=0.0014; *Supplementary file 1*). This indicates *r4>UAS-sun-RNAi* females have reduced nutrient-dependent body size plasticity, a finding that cannot be explained by changes to feeding

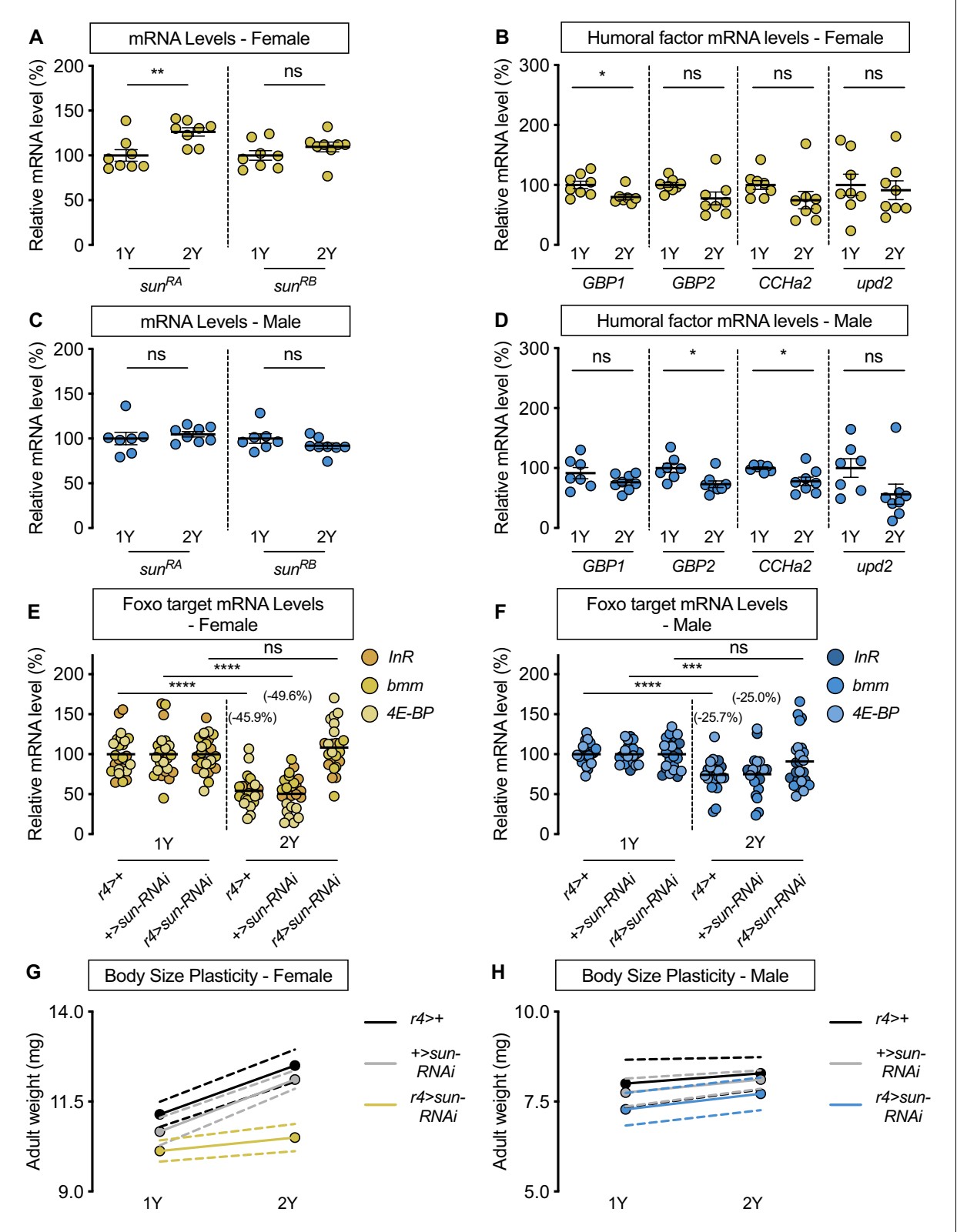

**Figure 3.** *stunted* is required for the nutrient-dependent upregulation of insulin pathway activity and increased female body size plasticity. (A) In females, mRNA levels of *stunted* (*sun*)^RA, but not *sun*^RB, were significantly higher in larvae cultured on a protein-rich diet (2Y) compared with larvae raised on a diet containing half the protein (1Y) (p=0.0055 and p=0.2327, respectively; Student's *t* test). n = 8 biological replicates. (B) mRNA levels of *Growth-blocking peptide 1* (*Gbp1*) were significantly different in females cultured on 2Y compared with females raised in 1Y (p=0.0245; Student's *t* test); *Figure 3 continued on next page*

*Figure 3 continued*

however, mRNA levels of *Growth-blocking peptide 2* (*Gbp2*), *CCHamide-2* (*CCHa2*), and *unpaired 2* (*upd2*) were not significantly different between female larvae raised on 1Y and 2Y (p=0.0662, 0.1416, and 0.7171, respectively; Student's *t* test). n = 7–8 biological replicates. (C) In males, mRNA levels of *sun*^RA and *sun*^RB were not significantly different in larvae raised on 2Y compared with larvae raised on 1Y (p=0.5832 and p=0.2017, respectively; Student's *t* test). n = 7–8 biological replicates. (D) Levels of *Gbp1* and *upd2* were not significantly different between male larvae raised on 2Y compared with larvae reared on 1Y (p=0.1487, and p=0.1686, respectively; Student's *t* test); whereas levels of *Gbp2* and *CCHa2* were significantly different between males raised in 2Y and 1Y (p=0.0214, and p=0.0272, respectively; Student's *t* test). n = 7–8 biological replicates. (E) In control *r4>+*, and *+>sun-RNAi* females, mRNA levels of Foxo targets (*insulin receptor* (*InR*), *brummer* (*bmm*), and *eukaryotic initiation factor 4E-binding protein* (*4E-BP*)), were significantly lower in larvae cultured on 2Y compared with larvae raised on 1Y (p<0.0001, for both comparisons; Student's *t* test). However, in *r4>sun-RNAi* females, there was no significant difference in Foxo target mRNA levels (p=0.2792; Student's *t* test). n = 8 biological replicates. (F) In control *r4>+*, and *+>sun-RNAi* males, mRNA levels of Foxo targets were significantly lower in larvae cultured on 2Y compared with larvae raised on 1Y (p<0.0001 and p=0.0001, respectively; Student's *t* test). While *r4>sun-RNAi* males showed no significant difference in Foxo target mRNA levels (p=0.2469; Student's *t* test), there was no genotype:diet interaction among males (p=0.1068), suggesting that genotype had no impact on Foxo target genes. Importantly, there was a significant sex:diet interaction for Foxo target mRNA levels in both the *r4>+* control (p=0.0166; two-way ANOVA followed by Tukey HSD test) and *+>sun-RNAi* control (p=0.0119; two-way ANOVA followed by Tukey HSD test), but not in *r4>sun-RNAi* larvae (p=0.1121; two-way ANOVA followed by Tukey HSD test). n = 7–8 biological replicates. (G) Adult weight was significantly higher in female flies raised in 2Y compared with females raised in 1Y in *r4>+* and *+>UAS-sun-RNAi* controls (p<0.0001 for both genotypes; two-way ANOVA followed by Tukey HSD test); however, adult weight was not significantly different between *r4>UAS-sun-RNAi* females reared on 2Y compared with genotype-matched females raised on 1Y (p=0.5035; two-way ANOVA followed by Tukey HSD test). n = 7–10 groups of 10 flies. (H) Adult weight was not significantly higher in male flies reared in 2Y compared with males cultured in 1Y for *r4>+* and *+>UAS-sun-RNAi* controls or *r4>UAS-sun-RNAi* males (p=0.8883, 0.6317, and 0.554, respectively; two-way ANOVA followed by Tukey HSD test). There was a significant sex:diet interaction in the *r4>+* and *+>UAS-sun-RNAi* control genotypes (p=0.011 and p=0.0005, respectively; two-way ANOVA followed by Tukey HSD test), but no sex:diet interaction in the *r4>UAS-sun-RNAi* genotype (p=0.8749; two-way ANOVA followed by Tukey HSD test). n = 6–9 groups of 10 flies. For all body size plasticity graphs, filled circles indicate mean body size, and dashed lines indicate 95% confidence interval. * indicates p<0.05, ** indicates p<0.01, *** indicates p<0.001 **** indicates p<0.0001; ns indicates not significant; error bars indicate SEM.

The online version of this article includes the following figure supplement(s) for figure 3:

**Figure supplement 1.** Increased circulating levels of Stunted (Sun) in females.
**Figure supplement 2.** Validation of *stunted* (*sun*) knockdown.
**Figure supplement 3.** No sex difference in food intake in fat body *stunted* (*sun*) knockdown larvae.
**Figure supplement 4.** Nutrient-dependent increased female body size plasticity requires *stunted* (*sun*).
**Figure supplement 5.** *methuselah* (*mth*) is dispensable for nutrient-dependent increased female body size plasticity.
**Figure supplement 6.** Most humoral factors have non-sex-specific effects on body size.
**Figure supplement 7.** *stunted* (*sun*) overexpression augments body size but does not confer increased body size plasticity in males.
**Figure supplement 8.** *stunted* (*sun*) overexpression augments body size in the diet used in *Delanoue et al., 2016* in males.

behavior (*Figure 3—figure supplement 3A*). In *r4>+*, *+>UAS-sun-RNAi*, and *r4>UAS-sun-RNAi* male flies raised on 2Y, adult weight was not significantly higher than in genotype-matched males raised on 1Y (*Figure 3H*; genotype:diet interaction p=0.9278; *Supplementary file 1*). Importantly, in contrast to the sex difference in nutrient-dependent body size plasticity we observed in the *r4>+* and *+>UAS-sun-RNAi* control genotypes (sex:diet interaction p=0.011 and p=0.0005, respectively; *Supplementary file 1*), the sex difference in phenotypic plasticity was abolished in the *r4>UAS-sun-RNAi* genotype (sex:diet interaction p=0.8749; *Supplementary file 1*), findings we reproduced using pupal volume (*Figure 3—figure supplement 4A,B*). While we observed no phenotypic plasticity effects in larvae with whole-body, pan-neuronal, or IPC loss of Sun receptor *methuselah* (*mth*; Fbgn0023000; *Delanoue et al., 2016*; *Figure 3—figure supplement 5A–F*), likely due to use of different *dilp2-GAL4* lines, minor variation in rearing conditions, and sex-specific plasticity defects in the *dilp2-GAL4* strain, we reproduced the female-specific effects of *sun* knockdown on body size using an additional fat body GAL4 line (*Figure 3—figure supplement 6A*). Further, we show that this role for *sun* in mediating the nutrient-dependent increase in female body size in a protein-rich context is unique to *sun*, as no other humoral factors caused sex-specific effects on body size (*Figure 3—figure supplement 6B,C*).

Our data suggests a model in which the nutrient-dependent increase in *sun* mRNA levels is one important reason that females raised in a protein-rich context have a larger body size. To determine whether increased *sun* mRNA levels could augment body size, we overexpressed *sun* specifically in the fat body in larvae of each sex reared on 1Y and 2Y. We found that fat body *sun* overexpression was sufficient to increase body size in both sexes, in both the 1Y and 2Y diets (*Figure 3—figure supplement 7A,B*). This demonstrates that increased *sun* mRNA levels are sufficient to enhance body

size in these contexts. While this finding contrasts with data from a previous study using a different diet and a mixed-sex experimental group (*Delanoue et al., 2016*), when we replicated their experimental conditions we found a significant increase in body size that was obscured by pooling data from males and females (*Figure 3—figure supplement 8A,B*). Together, this data supports a model in which increased fat body *sun* mRNA levels enhance body size in multiple nutritional contexts, an effect that was previously overlooked due to minor variation between lab diets and use of a mixed-sex experimental group. It is important to note, however, that despite the larger body size of *sun*-overexpressing males and females, phenotypic plasticity was not increased in the *sun*-overexpressing larvae (*Figure 3—figure supplement 7A,B*; diet:genotype interaction p=0.4959; and p=0.0895, respectively; *Supplementary file 1*). This is likely due to the fact that the nutrient-dependent increase in *sun* mRNA levels was still absent in the context of *sun* overexpression in males (*Figure 3—figure supplement 8C*), as our model suggests it is the ability to upregulate *sun* mRNA in response to dietary protein, rather than absolute *sun* mRNA levels, that allows females raised on a protein-rich diet to achieve a larger body size.

## Sex determination gene *transformer* promotes nutrient-dependent body size plasticity in females

To gain a more complete understanding of the sex difference in phenotypic plasticity, we wanted to identify genetic factors in females that confer the ability to upregulate *sun* mRNA levels in response to dietary protein. One candidate was sex determination gene *tra*, as *tra* was previously found to impact IIS activity and body size in a diet equivalent to 2Y (*Rideout et al., 2015*; *Mathews et al., 2017*). Thus, we performed loss- and gain-of-function studies with *tra* and monitored changes to IIS activity, *sun* mRNA, and body size in both the 1Y and 2Y diets. In control $w^{1118}$ females, Foxo target gene expression was significantly lower in larvae raised on 2Y compared with larvae cultured on 1Y (*Figure 4A*); however, this nutrient-dependent decrease in Foxo target gene expression was abolished in *tra* mutant females ($tra^{1}/Df(3L)st-j7$) (*Figure 4A*; diet:genotype interaction p=0.0081; *Supplementary file 1*). Similarly, while *sun* mRNA levels in $w^{1118}$ control females were significantly higher in larvae raised on 2Y compared with 1Y (*Figure 4B*), this nutrient-dependent increase in *sun* mRNA levels was absent in *tra* mutant females (*Figure 4B*). This indicates that *tra* is required in females for the nutrient-dependent increase in *sun* mRNA levels and IIS activity in a protein-rich context.

To determine whether lack of *tra* also impacts nutrient-dependent body size plasticity, we measured body size in $w^{1118}$ controls and *tra* mutants raised in 1Y and 2Y. In control $w^{1118}$ females, adult weight was significantly higher in flies raised on 2Y compared with flies cultured on 1Y (*Figure 4C*); however, this nutrient-dependent increase in adult weight was blocked in *tra* mutant females (*Figure 4C*; genotype:diet interaction p<0.0001; *Supplementary file 1*), a finding we reproduced using pupal volume (*Figure 4—figure supplement 1A*). Given that we confirmed this result using an additional *tra* mutant allele ($tra^{KO}$) (*Hudry et al., 2016*; *Figure 4—figure supplement 1B*), and that this finding cannot be explained by changes to food intake (*Figure 4—figure supplement 1C*), this indicates that *tra* mutant females have reduced nutrient-dependent body size plasticity compared with control females (genotype:diet interaction p<0.0001 [$tra^{1}/Df(3L)st-j7$]; p<0.0001 [$tra^{KO}$]; *Supplementary file 1*). In control $w^{1118}$ and *tra* mutant males, adult weight was not significantly higher in flies raised on 2Y compared with genotype-matched flies reared on 1Y (*Figure 4D*; genotype:diet interaction p=0.4507, *Supplementary file 1*). Given that we observed a sex difference in nutrient-dependent body size plasticity in the $w^{1118}$ genotype (sex:diet interaction p<0.0001; *Supplementary file 1*), but not in the *tra* mutant strains (sex:diet interaction p=0.6598 [$tra^{1}/Df(3L)st-j7$]; p=0.5068 [$tra^{KO}$]; *Supplementary file 1*), findings we replicated with pupal volume (*Figure 4—figure supplement 1D,E*), our data reveals a previously unrecognized requirement for *tra* in regulating the sex difference in nutrient-dependent phenotypic plasticity. To determine whether *tra* affects phenotypic plasticity via regulation of *sun*, we overexpressed *sun* in the fat body of *tra* mutant females. We found that the reduced body size of *tra* mutant females in 2Y was rescued by fat body *sun* overexpression (*Figure 4—figure supplement 2A*). This supports a model in which the smaller body size of *tra* mutant females reared in 2Y was due at least in part to lower *sun* mRNA levels.

To determine whether lack of a functional Tra protein in males explains their reduced nutrient-dependent body size plasticity, we overexpressed *UAS-tra^F* in all tissues using *daughterless* (*da*)-GAL4. We first asked whether Tra overexpression impacted the nutrient-dependent regulation of

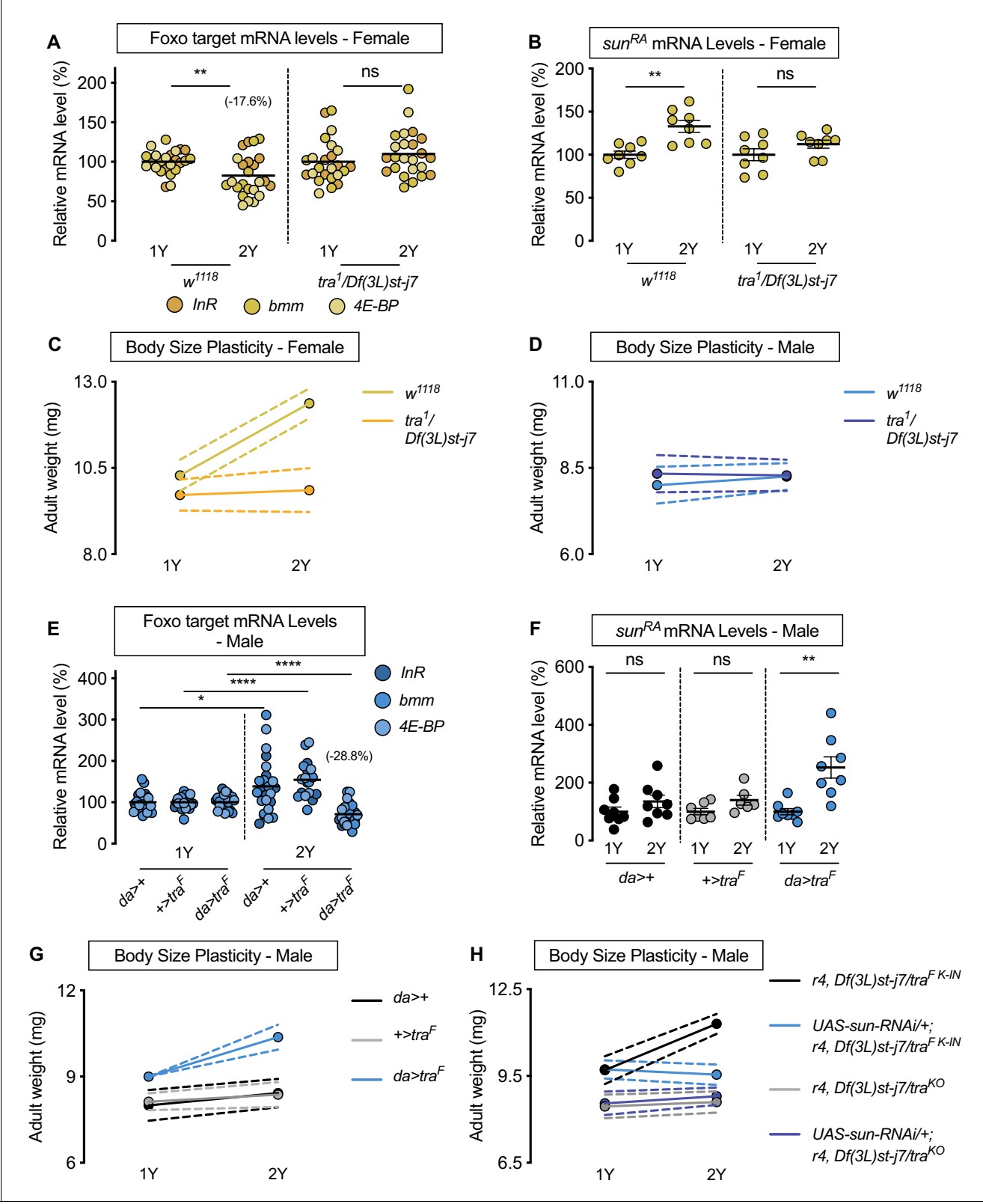

**Figure 4.** Sex determination gene *transformer* (*tra*) regulates increased nutrient-dependent body size plasticity in females. (**A**) In control *w1118* females, mRNA levels of Foxo targets (*insulin receptor* (*InR*), *brummer* (*bmm*), and *eukaryotic initiation factor 4E-binding protein* (*4E-BP*)), were significantly lower in larvae cultured on a protein-rich diet (2Y) compared with larvae raised on a diet containing half the protein (1Y) (p=0.0057; Student's *t* test). In *tra* mutant (*tra1/Df(3L)st-j7*) females, there was no significant difference in mRNA levels of Foxo targets in larvae cultured on 2Y compared with larvae raised

*Figure 4 continued on next page*

*Figure 4 continued*

on 1Y (p=0.2291 Student's *t* test). n = 8 biological replicates. (B) In control females, mRNA levels of *sun*$^{RA}$ were significantly higher in larvae cultured on 2Y compared with larvae raised on 1Y (p=0.0011; Student's *t* test); however, in *tra*$^{1}$/*Df(3L)st-j7* females there was no significant difference in *sun*$^{RA}$ mRNA levels between larvae cultured on 2Y compared with larvae raised on 1Y (p=0.1644; Student's *t* test). n = 8 biological replicates. (C) Adult weight was significantly higher in *w*$^{1118}$ females raised on 2Y compared with females reared on 1Y (p<0.0001; two-way ANOVA followed by Tukey HSD test); however, there was no significant difference in adult weight between *tra*$^{1}$/*Df(3L)st-j7* females cultured on 2Y compared with genotype-matched females raised on 1Y (p=0.9617; two-way ANOVA followed by Tukey HSD test). n = 7–8 groups of 10 flies. (D) Adult weight was not significantly higher in either *w*$^{1118}$ control or *tra*$^{1}$/*Df(3L)st-j7* mutant males in flies raised on 2Y compared with males reared on 1Y (p=0.7808 and p=0.9983, respectively; two-way ANOVA followed by Tukey HSD test). There was a significant sex:diet interaction in the *w*$^{1118}$ control genotype (p<0.0001; two-way ANOVA followed by Tukey HSD test); however, there was no sex:diet interaction in the *tra*$^{1}$/*Df(3L)st-j7* genotype (p=0.6598; two-way ANOVA followed by Tukey HSD test). n = 6–8 groups of 10 flies. (E) In control *da>+*, and *+>tra*$^{F}$ males, mRNA levels of Foxo targets were significantly higher in larvae cultured on 2Y compared with larvae raised on 1Y a diet containing half the protein content (1Y) (p=0.0108 and p<0.0001, respectively; Student's *t* test). However, in *da>tra*$^{F}$ males, there was a significant decrease in Foxo target mRNA levels (p<0.0001; Student's *t* test). n = 8 biological replicates. Importantly, there was a significant sex:diet interaction for Foxo target mRNA levels in both the *da>+* control (p=0.0004; two-way ANOVA followed by Tukey HSD test) and *+>tra*$^{F}$ control (p<0.0001; two-way ANOVA followed by Tukey HSD test), but not in *da>tra*$^{F}$ larvae (p=0.3095; two-way ANOVA followed by Tukey HSD test). n = 7–8 biological replicates. (F) In control *da>+* and *+>UAS-tra*$^{F}$ males, mRNA levels of *sun*$^{RA}$ were not significantly different between larvae cultured on 2Y compared with larvae raised on 1Y (p=0.2064 and p=0.0711, respectively; Student's *t* test). In contrast, *da>UAS-tra*$^{F}$ males showed a significant increase in mRNA levels of *sun*$^{RA}$ in larvae cultured on 2Y compared with males raised on 1Y (p=0.0013; Student's *t* test). n = 6–8 biological replicates. (G) Adult weight was not significantly higher in *da>+* and *+>UAS-tra*$^{F}$ control males reared on 2Y compared with genotype-matched males flies cultured on 1Y (p=0.5186 and p=0.8858, respectively; two-way ANOVA followed by Tukey HSD test); however, there was a significant increase in adult weight between *da>UAS-tra*$^{F}$ males cultured on 2Y compared with genotype-matched flies raised on 1Y (p<0.0001; two-way ANOVA followed by Tukey HSD test). n = 7–8 groups of 10 flies. (H) Adult weight was significantly higher in *r4-GAL4* control males with *tra*$^{F\ K-IN}$, which express physiological levels of a functional Tra protein, when reared on 2Y compared with 1Y ((p<0.0001 [*r4,Df(3L)st-j7/tra*$^{F\ K-IN}$]); two-way ANOVA followed by Tukey HSD test). In contrast, the nutrient-dependent increase in adult weight was abolished upon fat body knockdown of *sun* in a *tra*$^{F\ K-IN}$ male ((p=0.9915 [*UAS-sun-RNAi/+;r4,Df(3L)st-j7/tra*$^{F\ K-IN}$]); two-way ANOVA followed by Tukey HSD test). Adult weight was not different in *tra* mutant *r4-GAL4* males (*r4,Df(3L)st-j7/tra*$^{KO}$) reared on 2Y compared with genotype-matched males cultured on 1Y (p=0.9980; two-way ANOVA followed by Tukey HSD test). Adult weight was not reduced in 1Y with fat body knockdown of *sun* in a *tra* mutant male ((*UAS-sun-RNAi/+;r4,Df(3L)st-j7/tra*$^{KO}$) (p=0.9998 [*UAS-sun-RNAi/+; r4,Df(3L)st-j7/tra*$^{KO}$ v *r4,Df(3L)st-j7/tra*$^{KO}$]); two-way ANOVA followed by Tukey HSD test). n = 9–11 groups of 10 flies. For all body size plasticity graphs, filled circles indicate mean body size, and dashed lines indicate 95% confidence interval. * indicates p<0.05, ** indicates p<0.01, **** indicates p<0.0001; ns indicates not significant; error bars indicate SEM.

The online version of this article includes the following figure supplement(s) for figure 4:

**Figure supplement 1.** Increased nutrient-dependent body size plasticity in females requires *transformer*.

**Figure supplement 2.** Fat body *stunted* (*sun*) overexpression is sufficient to rescue the reduced body size of *transformer* (*tra*) mutant females in a protein-rich (2Y) diet.

**Figure supplement 3.** Sex determination gene *transformer* (*tra*) regulates increased nutrient-dependent body size plasticity.

*sun* mRNA and IIS activity. In control *da>+* and *+>UAS-tra*$^{F}$ males, there was no significant decrease in Foxo target gene expression in larvae reared in 2Y compared with larvae raised in 1Y (*Figure 4E*). In *da>UAS-tra*$^{F}$ males, however, there was a significant nutrient-dependent decrease in mRNA levels of Foxo target genes (*Figure 4E*). Because we observed a significant diet:genotype interaction (p<0.0001; *Supplementary file 1*), the magnitude of the nutrient-dependent increase in IIS activity in the *da>UAS-tra*$^{F}$ genotype was larger than in control males. Similarly, while *sun* mRNA levels in control *da>+* and *+>UAS-tra*$^{F}$ males were not significantly different in larvae raised on 2Y compared with larvae reared on 1Y (*Figure 4F*), there was a nutrient-dependent increase in *sun* mRNA levels in *da>UAS-tra*$^{F}$ males (*Figure 4F*). In *da>+*, *+>UAS-tra*$^{F}$, and *da>UAS-tra*$^{F}$ females, we observed a significant decrease in Foxo target gene expression, and a significant increase in *sun* mRNA levels (*Figure 4—figure supplement 3A,B*). Thus, the presence of a functional Tra protein in males confers the ability to upregulate *sun* mRNA levels and IIS activity, revealing that the lack of Tra in normal males accounts for the lack of a nutrient-dependent increase in *sun* mRNA and IIS activity.

We next tested whether the presence of a functional Tra protein in males would augment nutrient-dependent body size plasticity. We observed a significant increase in adult weight between *da>UAS-tra*$^{F}$ males reared on 2Y compared with genotype-matched males raised on 1Y (*Figure 4G*; genotype:diet interaction p=0.0038; *Supplementary file 1*). This nutrient-dependent increase was not present in either control *da>+* or *+>UAS-tra*$^{F}$ males (*Figure 4G*), a finding we reproduced using pupal volume (*Figure 4—figure supplement 3C*). Because one study suggested high levels of Tra expression may cause lethality (*Siera and Cline, 2008*), we repeated the experiment using males from a recently published strain of flies in which flies carry a cDNA encoding the female-specific Tra

protein knocked into the *tra* locus (*tra^F K-IN*). These males express Tra at a physiological level (*Hudry et al., 2019*). As with *da>UAS-tra^F* males, we found *tra^F K-IN* males had increased nutrient-dependent body size plasticity compared with control *w^1118* males and *tra^KO* males (*Figure 4—figure supplement 3D*; genotype:diet interaction p<0.0001; *Supplementary file 1*). Thus, males expressing a functional Tra protein have increased phenotypic plasticity compared with control males, revealing a new role for *tra* in conferring the ability to adjust body size in response to a protein-rich diet. In females, we observed a significant increase in both adult weight and pupal volume in *da>+*, *+>UAS-tra^F*, and *da>UAS-tra^F* flies raised on the 2Y diet compared with genotype-matched females cultured on the 1Y diet (*Figure 4—figure supplement 3E,F*); however, lack of a significant genotype:diet interaction indicates that phenotypic plasticity in *da>UAS-tra^F* females was not different from controls (p=0.5912; *Supplementary file 1*), findings we reproduced with the *tra^F K-IN* allele (*Figure 4—figure supplement 3G*; genotype:diet interaction p<0.0001 *Supplementary file 1*). Importantly, the sex difference in nutrient-dependent body size plasticity that we observed in the *w^1118* control genotype (sex:diet interaction p<0.0001; *Supplementary file 1*) was abolished between *tra^F K-IN* males and their genotype-matched females (p=0.3168; *Supplementary file 1*). To determine whether the nutrient-dependent upregulation of *sun* mRNA was required for Tra to enhance male body size in a protein-rich context, we overexpressed the *UAS-sun-RNAi* transgene in the fat body of *tra^F K-IN* males. We found that the nutrient-dependent body size increase in *tra^F K-IN* males was blocked in males with fat body *sun* loss (*Figure 4H*), a finding we reproduced in *tra^F K-IN* females (*Figure 4—figure supplement 3H*). This indicates that the nutrient-dependent upregulation of *sun* mRNA in larvae with a functional Tra protein is required for phenotypic plasticity. Together, these data demonstrate a new role for Tra in regulating the sex difference in nutrient-dependent body size plasticity, and identify fat body *sun* as one downstream factor that mediates Tra's effects on phenotypic plasticity.

## Transcriptional coactivator *spargel* represents one link between transformer and regulation of *sun* mRNA levels

While sex determination gene *tra* impacts sexual differentiation via regulation of confirmed target genes *doublesex* (*dsx*; FBgn0000504) and *fruitless* (*fru*; FBgn0004652), neither *dsx* nor *fru* affect body size (*Rideout et al., 2015*). Given the key role of *sun* in mediating the nutrient-dependent increase in body size downstream of Tra, we wanted to identify the link between Tra and regulation of *sun* mRNA levels. Previous studies show that transcriptional coactivator *spargel* (*srl*, FBgn0037248), the *Drosophila* homolog of *peroxisome proliferator-activated receptor gamma coactivator 1-alpha* (*PGC-1α*) (*Tiefenböck et al., 2010*), coordinates *sun* mRNA levels with dietary protein (*Delanoue et al., 2016*). To test whether Srl mediates the sex difference in nutrient-dependent upregulation of *sun* mRNA levels, we examined mRNA levels of *sun* in female larvae heterozygous for a strong hypomorphic allele of *srl* (*srl^1/+*) (*Tiefenböck et al., 2010*). In females, we found that the nutrient-dependent upregulation of *sun* mRNA levels in *w^1118* control larvae was blunted in *srl^1/+* larvae (*Figure 5A*; diet:genotype interaction p<0.0001; *Supplementary file 1*). To determine whether a smaller nutrient-dependent increase in *sun* mRNA levels affects the ability of *srl^1/+* larvae to achieve a larger body size in a protein-rich context, we raised *srl^1/+* larvae on 1Y and 2Y. While we confirmed that *srl^1/+* larvae have no generalized developmental defects, as there was no decrease in body size in *srl^1/+* female or male larvae reared on 1Y (*Figure 5B,C*), we showed that the nutrient-dependent increase in body size in *srl^1/+* females was eliminated (*Figure 5B*). Given that adult weight was significantly higher in control *w^1118* females raised on 2Y compared with genotype-matched females cultured on 1Y (*Figure 5B*), this indicates that *srl^1/+* females have reduced nutrient-dependent body size plasticity (genotype:diet interaction p<0.0001; *Supplementary file 1*). In control *w^1118* and *srl^1/+* males, adult weight was not significantly higher in flies raised on 2Y compared with genotype-matched flies reared on 1Y (*Figure 5C*; genotype:diet interaction p=0.8323). This result suggests that Srl mediates the nutrient-dependent upregulation of *sun* mRNA levels and increased nutrient-dependent body size plasticity in female larvae in a protein-rich context, where future studies will need to determine whether Srl also impacts the sex difference in circulating Sun. Indeed, while Sun is also regulated at the level of secretion by fat body Target-of-Rapamycin (TOR) signaling (*Delanoue et al., 2016*), we found no sex difference in fat body TOR activity in either 1Y or 2Y (*Figure 5—figure supplement 1A–D*). Given that TOR activity does not affect *sun* mRNA levels (*Delanoue et al., 2016*), which we confirm (*Figure 5—figure supplement 1E*), our data indicates

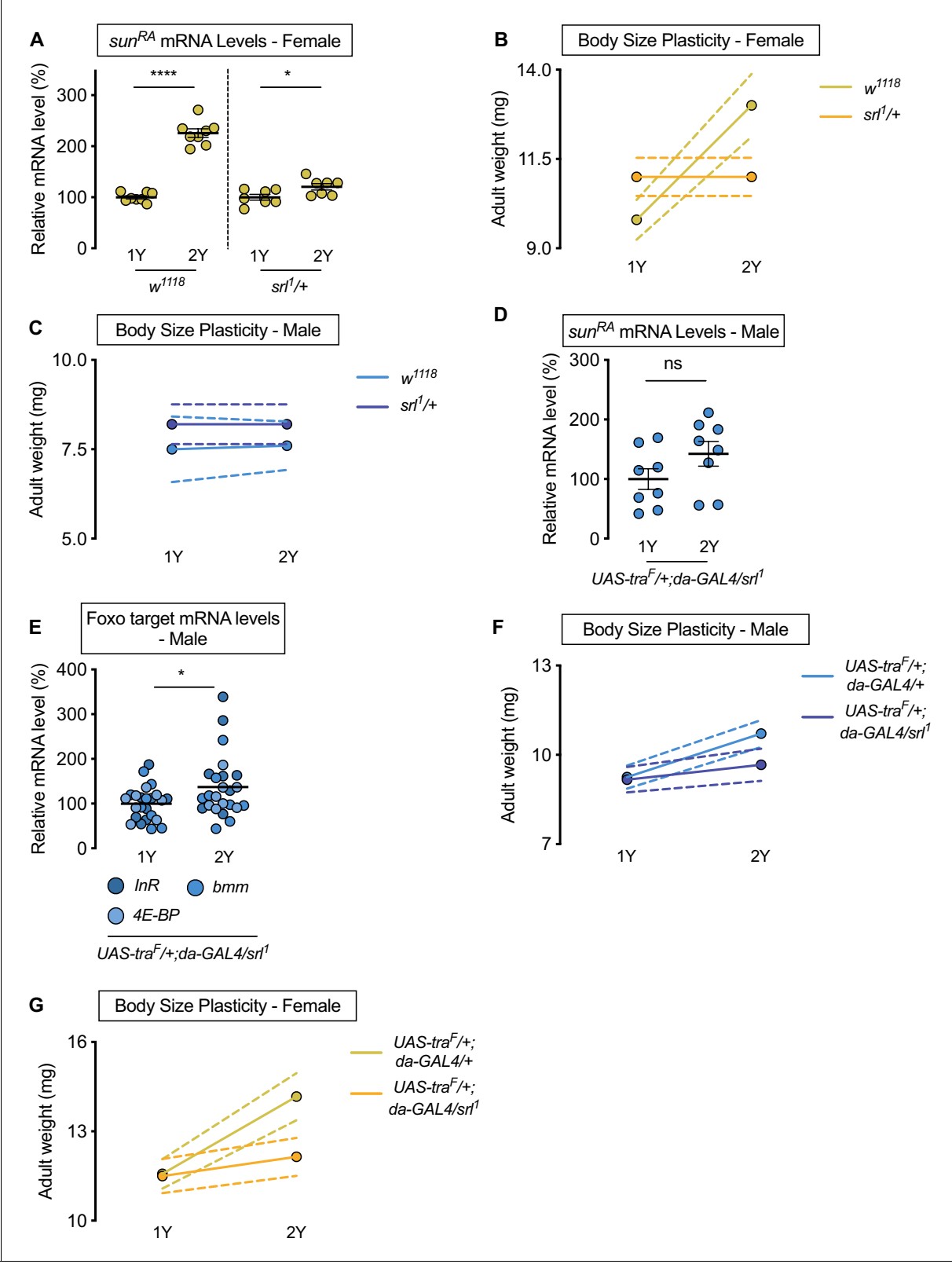

**Figure 5.** Sex determination gene *transformer* (*tra*) requires transcriptional coactivator *spargel* (*srl*) for increased nutrient-dependent body size plasticity in females. (A) In control *w1118* females and females with heterozygous loss of *srl* (*srl1/+*), mRNA levels of *sunRA* were significantly higher in larvae cultured on a protein-rich diet (2Y) compared with larvae raised on a diet with half the protein (1Y) (p<0.0001 and p=0.0301; Student's *t* test); however, there was a significant genotype:diet interaction indicating that the protein-dependent upregulation of *sunRA* was blunted in *srl1/+* females (p<0.0001;

*Figure 5 continued on next page*

*Figure 5 continued*

two-way ANOVA followed by Tukey HSD test). n = 7–8 biological replicates. (**B**) Adult weight was significantly higher in $w^{1118}$ females raised on 2Y compared with females reared on 1Y (p<0.0001; two-way ANOVA followed by Tukey HSD test); however, there was no significant difference in adult weight between $srl^1$/+ females cultured on 2Y compared with genotype-matched females raised on 1Y (p>0.9999; two-way ANOVA followed by Tukey HSD test). n = 5–7 groups of 10 flies. (**C**) Adult weight was not significantly higher in either $w^{1118}$ control or $srl^1$/+ mutant males in flies raised on 2Y compared with males reared on 1Y (p=0.9906 and p>0.9999, respectively; two-way ANOVA followed by Tukey HSD test). n = 4–5 groups of 10 flies. (**D**) mRNA levels of $sun^{RA}$ were not significantly different in $da>tra^F$ males with heterozygous loss of $srl$ ($UAS$-$tra^F$/+;$da$-$GAL4$/$srl^1$) cultured on 1Y compared to genotype matched males cultured on 2Y (p=0.1405; Student's *t* test). n = 8 biological replicates. (**E**) In control $da>tra^F$ males with heterozygous loss of $srl$, mRNA levels of Foxo targets (*insulin receptor* (*InR*), *brummer* (*bmm*), and *eukaryotic initiation factor 4E-binding protein* (*4E-BP*)), were significantly higher in larvae cultured on 2Y compared with larvae raised on 1Y (p=0.0266; Student's *t* test). n = 8 biological replicates. (**F**) Adult weight was higher in $da>UAS$-$tra^F$ males raised on 2Y compared with $da>UAS$-$tra^F$ males reared on 1Y (p<0.0001; two-way ANOVA followed by Tukey HSD test). In contrast, the nutrient-dependent increase in adult weight was abolished in $da>UAS$-$tra^F$ males heterozygous for $srl^1$ (p=0.2811; two-way ANOVA followed by Tukey HSD test). n = 6–8 groups of 10 flies. (**G**) Adult weight was higher in $da>UAS$-$tra^F$ females raised on 2Y compared with $da>UAS$-$tra^F$ females reared on 1Y (p<0.0001; two-way ANOVA followed by Tukey HSD test). In contrast, the nutrient-dependent increase in adult weight was absent in $da>UAS$-$tra^F$ females heterozygous for $srl^1$ (p=0.2927; two-way ANOVA followed by Tukey HSD test). n = 6–7 groups of 10 flies. For all body size plasticity graphs, filled circles indicate mean body size, and dashed lines indicate 95% confidence interval. * indicates p<0.05, **** indicates p<0.0001; ns indicates not significant; error bars indicate SEM.

The online version of this article includes the following figure supplement(s) for figure 5:

**Figure supplement 1.** No nutrient- or transformer-dependent sex difference in fat body Target-of-rapamycin (TOR) signaling activity.

**Figure supplement 2.** *transformer* (*tra*) is required for nutrient-dependent upregulation of *spargel* (*srl*) target expression in females, but not all *srl* targets are not required for increased female nutrient-dependent body size plasticity.

that the sex difference in nutrient-dependent upregulation of *sun* mRNA levels is due to Srl, and not TOR. This aligns with our previous finding that treating larvae with TOR inhibitor Rapamycin did not cause sex-biased effects on larval growth (*Rideout et al., 2015*).

To determine whether Srl mediates the Tra-dependent regulation of *sun* mRNA levels, we measured *sun* mRNA levels in males with ectopic Tra expression ($da>UAS$-$tra^F$). While $da>UAS$-$tra^F$ males show a significant nutrient-dependent upregulation of *sun* mRNA levels compared with $da>+$ and $+>UAS$-$tra^F$ control males (*Figure 4F*), we found that *sun* mRNA levels were no longer higher in $da>UAS$-$tra^F$ males heterozygous for the $srl^1$ allele raised on 2Y compared with genotype-matched males reared on 1Y (*Figure 5D*). Similarly, we observed no decrease in Foxo target genes between $da>UAS$-$tra^F$ males heterozygous for the $srl^1$ allele raised on 2Y compared with genotype-matched males reared on 1Y (*Figure 5E*), indicating the nutrient-dependent upregulation of IIS activity in $da>UAS$-$tra^F$ males was abolished in the context of reduced Srl function. Given that we observed no Tra-dependent changes to TOR activity (*Figure 5—figure supplement 1F,G*), when taken together our data indicates that Srl function is required for the Tra-dependent increase in *sun* mRNA levels in a protein-rich context. Srl therefore represents an additional link between sex determination gene *tra* and the regulation of gene expression. Moreover, we show that the Srl-dependent regulation of *sun* downstream of Tra is significant for phenotypic plasticity, as the nutrient-dependent increase in body size was blocked in $da>UAS$-$tra^F$ females and males heterozygous for the $srl^1$ allele (*Figure 5F, G*; genotype:diet interaction p=0.0146 and p=0.0008, respectively). While we find that Srl targets other than *sun* were also regulated in a sex-specific manner by nutrients and Tra function (*Figure 5—figure supplement 2A–D*), other functionally similar Srl targets did not reproduce sex-specific changes to nutrient-dependent body size plasticity that we observed upon loss of fat body *sun* (*Figure 5—figure supplement 2E–H*). Although we cannot rule out all Srl targets, our data indicates a key role for *sun* among Srl targets in mediating the effects of Tra on nutrient-dependent body size plasticity. This reveals a previously unrecognized role for Srl in mediating sex-specific changes to gene expression, and identifies Srl as a new link between Tra and nutrient-dependent changes to gene expression.

## Increased nutrient-dependent body size plasticity in females promotes fecundity in a protein-rich context

Previous studies have shown that plentiful nutrients during development maximize body size to promote fertility in *Drosophila* females (*Bergland et al., 2008*; *Green and Extavour, 2014*; *Grönke et al., 2010*; *Hodin and Riddiford, 2000*; *Klepsatel et al., 2020*; *Mendes and Mirth,*

*2016*; *Robertson, 1957a*; *Robertson, 1957b*; *Sarikaya et al., 2012*; *Tu and Tatar, 2003*), and that high levels of IIS activity are required for normal egg development, ovariole number, and fecundity (*Green and Extavour, 2014*; *Grönke et al., 2010*; *Mendes and Mirth, 2016*; *Richard et al., 2005*). In line with these findings, $w^{1118}$ female flies reared on 2Y produced significantly more eggs compared with genotype-matched females cultured on 1Y (*Figure 6A*). This aligns with findings from many studies showing that increased nutrients promote fertility (*Green and Extavour, 2014*; *Grönke et al., 2010*; *Mendes and Mirth, 2016*; *Richard et al., 2005*) and suggests that the ability to augment IIS activity and body size in response to a protein-rich diet allows females to maximize fecundity in conditions where nutrients are plentiful. To test this, we measured the number of eggs produced by $InR^{E19}/+$ females and $w^{1118}$ controls raised in either 1Y or 2Y. In contrast to $w^{1118}$ females, the nutrient-dependent increase in egg production was absent in $InR^{E19}/+$ females (*Figure 6A*). Similarly, there was no diet-induced increase in egg production in *dilp2* mutant females (*Figure 6B*). These findings suggest that the nutrient-dependent increase in IIS activity and body size are important to promote fecundity in a protein-rich context. This result aligns with findings from a previous study showing that lifetime fecundity was significantly lower in *dilp2* mutants raised in a yeast-rich diet (*Grönke et al., 2010*). To extend our findings beyond *dilp* genes, we next examined fecundity in females with an RNAi-mediated reduction in *sun*. We found that the nutrient-dependent increase in egg production in *r4>UAS-sun-RNAi* females was eliminated, in contrast to the robust diet-induced increase in fecundity in *r4>+* and *+>UAS-sun-RNAi* control females (*Figure 6C*). Together, this data suggests that *dilp2* and fat body-derived *sun* play a role in maximizing IIS activity and body size to promote egg production in a protein-rich context. Future studies will need to determine which aspect of ovary development is affected by these genetic manipulations (*Green and Extavour, 2014*; *Grönke et al., 2010*; *Mendes and Mirth, 2016*; *Richard et al., 2005*), whether this phenotype is specific to *dilp2*, and whether the effects require *InR* function in the ovary or in other tissues.

In males, which have a reduced ability to augment body size in response to a protein-rich diet, we also investigated the relationship between nutrient content, body size, and fertility. When we compared fertility in $w^{1118}$ males reared on 1Y compared with males raised on 2Y, we found no significant difference in the number of offspring produced (*Figure 6D*). Thus, neither male body size nor fertility were enhanced by rearing flies in a protein-rich environment. Given that previous studies suggest that a larger body size in males promotes reproductive success (*Ewing, 1961*; *Partridge et al., 1987*; *Partridge and Farquhar, 1983*), we next asked whether genetic manipulations that augment male body size also increased fertility. One way to augment male body size in 1Y is heterozygous loss of *phosphatase and tensin homolog* (*pten*, FBgn0026379; $pten^{2L100}/+$) (*Figure 1—figure supplement 7B*). Interestingly, fertility was not significantly higher in $pten^{2L100}/+$ males compared with $w^{1118}$ controls raised in 1Y (*Figure 6D*), suggesting that a larger body size does not always augment fertility in males. Similarly, when we measured fertility in *r4>UAS-sun* males, which are larger than control males (*Figure 3—figure supplement 7B*), fertility was not significantly different from *r4>+* and *+>UAS-sun* control males (*Figure 6E*). Interestingly, when we examined fertility in $pten^{2L100}/+$ and *r4>UAS-sun* males in 2Y, fertility was significantly increased in $pten^{2L100}/+$ males compared with genotype-matched controls cultured in 1Y (*Figure 6D*), an observation we did not repeat in *r4>UAS-sun* males (*Figure 6E*). Ultimately, this less robust and more complex relationship between body size and fertility in males suggests a possible explanation for their decreased nutrient-dependent body size plasticity compared with females.

## Discussion

In many animals, body size plasticity in response to environmental factors such as nutrition differs between the sexes (*Fairbairn, 1997*). While past studies have identified mechanisms underlying nutrient-dependent growth in a mixed-sex population, and revealed factors that promote sex-specific growth in a single nutritional context, the mechanisms underlying the sex difference in nutrient-dependent body size plasticity remain unknown. In this study, we showed that females have higher phenotypic plasticity compared with males when reared on a protein-rich diet, and elucidated the molecular mechanisms underlying the sex difference in nutrient-dependent body size plasticity in this context. Our data suggests a model in which high levels of dietary protein augment female body size by stimulating an increase in IIS activity, where we identified a requirement for *dilp2* and

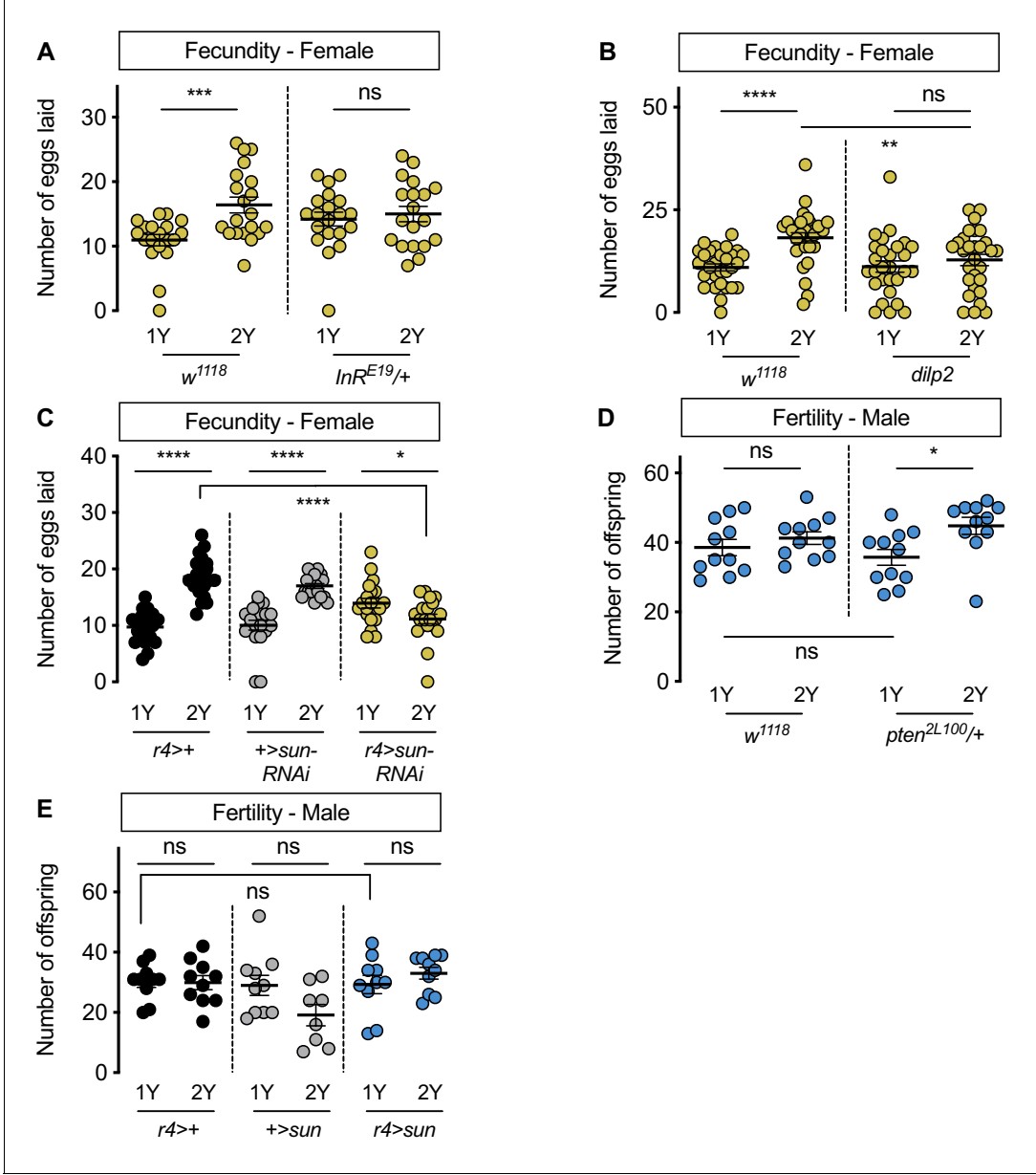

**Figure 6.** Increased nutrient-dependent body size plasticity in females promotes fertility. (A) In control $w^{1118}$ females, there was a significant increase in the number of eggs laid by females raised on a protein-rich diet (2Y) compared with females reared on a diet with half the protein (1Y) (p=0.0009; Student's *t* test); however, there was no significant difference in the number of eggs laid between $InR^{E19}/+$ females cultured on 2Y compared with genotype-matched females raised on 1Y (p=0.617; Student's *t* test). n = 19–20 biological replicates. (B) In control $w^{1118}$ females, there was a significant increase in the number of eggs laid by females raised on 2Y compared with females cultured on 1Y (p<0.0001; Student's *t* test); however, there was no significant difference in the number of eggs laid between *dilp2* mutant females cultured on 2Y compared with females raised on 1Y (p=0.4105; Student's *t* test). n = 28–30 biological replicates. (C) In control *r4>+* and *+>UAS-sun-RNAi* females, there was a significant increase in the number of eggs laid by females raised on 2Y compared with control females cultured on 1Y (p<0.0001 for both genotypes; Student's *t* test). In *r4>UAS-sun-RNAi* females, the number of eggs laid by females cultured on 2Y was lower than females raised on 1Y (p=0.0243; Student's *t* test). n = 20 biological replicates. (D) In control $w^{1118}$ males, there was no significant difference in the number of offspring produced between a 1Y and 2Y diet (p=0.3662; Student's *t* test). There was also no significant difference in the number of offspring produced between control $w^{1118}$ males and males heterozygous for a loss-of-function allele of *phosphatase and tensin homolog* (*pten*; genotype $pten^{2L100}/+$) raised on 1Y (p=0.4003; Student's *t* test). Unlike control males, $pten^{2L100}/+$ males reared on 2Y produced significantly more offspring than genotype-matched males raised on 1Y (p=0.0137; Student's *t* test). n = 11 biological replicates. (E) In control *r4>+* and *+>UAS-sun* and *r4>UAS-sun* males, there was no significant effect on the number of offspring produced between a 1Y and 2Y diet (p=0.9222, 0.0595, and 0.32 respectively; Student's *t* test). There was also no significant difference in the number of offspring produced between control *r4>+*, *+>UAS-sun* males and *r4>UAS-sun* males raised on 1Y (p=0.9723 and p=0.9969 respectively; one-way ANOVA

*Figure 6 continued on next page*

**Figure 6 continued**

followed by Tukey HSD test). n = 8–10 groups of 10 flies. * indicates p<0.05, ** indicates p<0.01, *** indicates p<0.001, **** indicates p<0.0001; ns indicates not significant; error bars indicate SEM.

---

*sun* in promoting this nutrient-dependent increase in IIS activity. Importantly, we discovered *tra* as the factor responsible for stimulating *sun* mRNA levels and IIS activity in a protein-rich context, revealing a novel role for sex determination gene *tra* in regulating phenotypic plasticity. Mechanistically, *tra* enhanced *sun* mRNA levels and body size in protein-rich conditions via transcriptional coactivator Srl, identifying Srl as one link between *tra* and the nutrient-dependent regulation of gene expression. Together, these findings provide new insight into how *Drosophila* females achieve increased nutrient-dependent body size plasticity compared with males.

One key feature of this increased phenotypic plasticity in females was a female-biased increase in IIS activity in a protein-rich context. This reveals a previously unrecognized sex difference in the coupling between IIS activity and dietary protein. In females, there was tight coupling between increased nutrient input and enhanced IIS activity across a wide protein concentration range in all control genotypes. In males, this close coordination between dietary protein and IIS activity was weaker in a protein-rich context. Our data shows that sex-biased nutrient-dependent change to IIS activity during development is physiologically significant, as it supports an increased rate of growth and consequently larger body size in females but not in males raised on a protein-rich diet. In future studies, it will be important to determine whether the sex difference in coupling between nutrients and IIS activity exists in other contexts. For example, previous studies on the extension of life span by dietary restriction have shown that male and female flies differ in the concentration of nutrients that produces the maximum life span extension, and in the magnitude of life span extension produced by dietary restriction (*Magwere et al., 2004*; *Regan et al., 2016*). Similar sex-specific effects of dietary restriction and reduced IIS on life span have also been observed in mice (*Holzenberger et al., 2003*; *Kane et al., 2018*; reviewed in *Regan and Partridge, 2013*; *Selman et al., 2008*) and humans (*Van Heemst et al., 2005*). Future studies will be needed to determine whether a male-female difference in coupling between nutrients and IIS activity account for these sex-specific life span responses to dietary restriction. Indeed, given that sex differences have been reported in the risk of developing diseases associated with overnutrition and dysregulation of IIS activity such as obesity and type 2 diabetes (*Kautzky-Willer et al., 2016*; *Mauvais-Jarvis, 2018*; *Tramunt et al., 2020*), more detailed knowledge of the male-female difference in coupling between nutrients and IIS activity in other models may provide insights into this sex-biased risk of disease.

In addition to revealing a sex difference in the nutrient-dependent upregulation of IIS activity, our data identified a female-specific requirement for *dilp2* and *sun* in mediating the diet-induced increase in IIS activity in a protein-rich context. While previous studies have shown that both *dilp2* and *sun* positively regulate body size (*Ikeya et al., 2002*; *Grönke et al., 2010*; *Delanoue et al., 2016*), we describe new sex-specific roles for *dilp2* and *sun* in nutrient-dependent phenotypic plasticity. Elegant studies have shown that *sun* is a secreted factor that stimulates Dilp2 release from the IPCs (*Delanoue et al., 2016*). Together with our data, this suggests a model in which females are able to achieve a larger body size in a protein-rich diet because they have the ability to upregulate *sun* mRNA levels, whereas males do not. Indeed, we show that higher *sun* mRNA levels are sufficient to augment body size. This model aligns well with findings from two previous studies on Dilp2 secretion in male and female larvae. The first study, which raised larvae on a protein-rich diet equivalent to the 2Y diet, found increased Dilp2 secretion in females compared to males (*Rideout et al., 2015*). The second study, which raised larvae on a diet equivalent to the 1Y diet, found no sex difference in Dilp2 secretion and no effects of *dilp2* loss on body size (*Sawala and Gould, 2017*). Thus, while these previous studies differed in their initial findings on a sex difference in Dilp2 secretion, our data reconcile these minor differences by identifying context-dependent effects of *dilp2* on body size. It is important to note that absolute confirmation of a sex difference in hemolymph Dilp2 levels will be needed in future studies because the body size plasticity defects in the *dilp2-HF* strain precluded its use as a tool to quantify circulating Dilp2 levels in our study. Future studies will also need to determine whether these sex-specific and context-dependent effects of *dilp2* are observed in other phenotypes regulated by *dilp2* and other *dilp* genes. For example, flies carrying mutations in *dilp* genes show changes to aging, metabolism, sleep, and immunity, among other phenotypes (*Bai et al.,*

*2012*; *Brown et al., 2020*; *Cong et al., 2015*; *Grönke et al., 2010*; *Liu et al., 2016*; *Nässel and Vanden Broeck, 2016*; *Okamoto et al., 2009*; *Okamoto and Nishimura, 2015*; *Post et al., 2018*; *Post et al., 2019*; *Slaidina et al., 2009*; *Stafford et al., 2012*; *Zhang et al., 2009*; *Brogiolo et al., 2001*; *Cognigni et al., 2011*; *Linneweber et al., 2014*; *Semaniuk et al., 2018*; *Suzawa et al., 2019*; *Ugrankar et al., 2018*). Further, it will be interesting to determine whether the sex-specific regulation of *sun* is observed in any other contexts, and whether it will influence sex differences in phenotypes associated with altered IIS activity, such as life span.

While our findings on *sun* and *dilp2* provide mechanistic insight into the molecular basis for the larger body size of females reared on a protein-rich diet, a key finding from our study was the identification of sex determination gene *tra* as the factor that confers plasticity to females. Normally, nutrient-dependent body size plasticity is higher in females than in males in a protein-rich context. In females lacking a functional Tra protein, however, this increased nutrient-dependent body size plasticity was abolished. In males, which normally lack a functional Tra protein, ectopic Tra expression conferred increased nutrient-dependent body size plasticity. While a previous study showed that on the 2Y diet Tra promotes Dilp2 secretion (*Rideout et al., 2015*), our current study extends this finding in two ways: by identifying *sun* as one link between Tra, Dilp2, and changes to IIS activity; and by showing that Tra regulates *sun* mRNA via conserved transcriptional coactivator Srl. While previous studies discovered Srl as the factor that promotes *sun* mRNA levels in response to dietary protein in a mixed-sex larval population (*Delanoue et al., 2016*), our findings reveal a previously unrecognized sex-specific role for Srl in regulating transcription. Because loss of Tra reduces Srl transcriptional activity, this new link between Tra and Srl suggests an additional way in which Tra may impact gene expression beyond its canonical downstream targets *dsx* and *fru*. While this builds on recent studies that reveal a number of additional Tra-regulated genes (*Clough et al., 2014*; *Hudry et al., 2016*; *Hudry et al., 2019*), it will be important to determine whether these additional Tra-regulated genes including *sun* represent direct targets of Tra/Srl. Future studies will also be needed to elucidate how Tra impacts Srl transcriptional activity in a context-dependent manner. However, uncovering a connection between a sex determination gene and a key regulator of genes involved in mitochondrial function suggests an additional mechanism that may contribute to sex differences in phenotypes affected by mitochondrial function (e.g. lifespan) (*Tiefenböck et al., 2010*; *Cho et al., 2011*; *Tower, 2015*; *Tower, 2017*). In addition, it will be critical to explore how the presence of Tra allows an individual to couple dietary protein with body size. Because the *tra* locus is regulated both by alternative splicing and transcription (*Belote et al., 1989*; *Boggs et al., 1987*; *Grmai et al., 2018*; *Inoue et al., 1990*; *Sosnowski et al., 1989*), and Tra protein is regulated by phosphorylation (*Du et al., 1998*), our study highlights the importance of additional studies on the regulation of the *tra* genomic locus and Tra protein throughout development to gain mechanistic insight into its effects on nutrient-dependent body size plasticity.

While the main outcome of our work was to reveal the molecular mechanisms that regulate the sex difference in nutrient-dependent body size plasticity, we also provide some insight into how genes that contribute to nutrient-dependent body size plasticity affect female fecundity and male fertility. Our findings align well with previous studies demonstrating that increased nutrient availability during development and a larger female body size confers increased ovariole number and fertility (*Green and Extavour, 2014*; *Klepsatel et al., 2020*; *Mendes and Mirth, 2016*; *Robertson, 1957a*; *Robertson, 1957b*), as females lacking either *dilp2* or fat body-derived *sun* were unable to augment egg production in a protein-rich context. Given that previous studies demonstrate IIS activity influences germline stem cells in the ovary in adult flies (*Hsu et al., 2008*; *Hsu and Drummond-Barbosa, 2009*; *Kao et al., 2015*; *LaFever and Drummond-Barbosa, 2005*; *Lin and Hsu, 2020*; *Su et al., 2018*), there is a clear reproductive benefit that arises from the tight coupling between nutrient availability, IIS activity, and body size in females. In males, however, the relationship between fertility and body size remains less clear. While larger males are more reproductively successful both in the wild and in laboratory conditions (*Ewing, 1961*; *Partridge and Farquhar, 1983*), other studies revealed that medium-sized males were more fertile than both larger and smaller males (*Lefranc and Bundgaard, 2000*). Given that our study revealed no significant increase in the number of progeny produced by larger males, the fertility benefits that accompany a larger body size in males may be context-dependent. For example, a larger body size increases the ability of males to outcompete smaller males (*Flatt, 2020*; *Partridge et al., 1987*; *Partridge and Farquhar, 1983*). Thus, in crowded situations, a bigger body may provide significant fertility gains. On the other hand,

in conditions where nutrients are limiting, an imbalance in the allocation of energy from food to growth rather than to reproduction may decrease fertility (*Bass et al., 2007*; *Camus et al., 2017*; *Jensen et al., 2015*; *Wood et al., 2018*). Future studies will need to resolve the relationship between body size and fertility in males, as this will suggest the ultimate reason(s) for the sex difference in nutrient-dependent body size plasticity.

# Materials and methods

## Key resources table

| Reagent type (species) or resource | Designation | Source or reference | Identifiers | Additional information |
|---|---|---|---|---|
| Genetic reagent (*Drosophila melanogaster*) | Canton-S | Bloomington *Drosophila* stock center | BDSC: 64349 | |
| Genetic reagent (*Drosophila melanogaster*) | $w^{1118}$ | Bloomington *Drosophila* stock center | BDSC: 3605 | |
| Genetic reagent (*Drosophila melanogaster*) | $tra^1$ | Bloomington *Drosophila* stock center | BDSC: 675 | |
| Genetic reagent (*Drosophila melanogaster*) | Df(3L)st-j7 | Bloomington *Drosophila* stock center | BDSC: 5416 | |
| Genetic reagent (*Drosophila melanogaster*) | $srl^1$ | Bloomington *Drosophila* stock center | BDSC: 14965 | |
| Genetic reagent (*Drosophila melanogaster*) | $InR^{E19}$ | Bloomington *Drosophila* stock center | BDSC: 9646 | |
| Genetic reagent (*Drosophila melanogaster*) | TRiP Control | Bloomington *Drosophila* stock center | BDSC: 36303 | |
| Genetic reagent (*Drosophila melanogaster*) | UAS-dilp2-RNAi | Bloomington *Drosophila* stock center | BDSC: 32475 | |
| Genetic reagent (*Drosophila melanogaster*) | UAS-upd2-RNAi | Bloomington *Drosophila* stock center | BDSC: 33949 | |
| Genetic reagent (*Drosophila melanogaster*) | $UAS-tra^F$ | Bloomington *Drosophila* stock center | BDSC: 4590 | |
| Genetic reagent (*Drosophila melanogaster*) | UAS-rheb | Bloomington *Drosophila* stock center | BDSC: 9688 | |
| Genetic reagent (*Drosophila melanogaster*) | UAS-cyt-c-p-RNAi | Bloomington *Drosophila* stock center | BDSC: 64898 | |
| Genetic reagent (*Drosophila melanogaster*) | UAS-Idh-RNAi | Bloomington *Drosophila* stock center | BDSC: 41708 | |
| Genetic reagent (*Drosophila melanogaster*) | $mth^1$ | Bloomington *Drosophila* stock center | BDSC: 27896 | |
| Genetic reagent (*Drosophila melanogaster*) | $y^1,w^1$ | Bloomington *Drosophila* stock center | BDSC: 1495 | |

*Continued on next page*

*Continued*

| Reagent type (species) or resource | Designation | Source or reference | Identifiers | Additional information |
|---|---|---|---|---|
| Genetic reagent (*Drosophila melanogaster*) | UAS-sun-RNAi | Vienna *Drosophila* resource center | VDRC: GD23685 | |
| Genetic reagent (*Drosophila melanogaster*) | UAS-Gbp1-RNAi | Vienna *Drosophila* resource center | VDRC: KK108755 | |
| Genetic reagent (*Drosophila melanogaster*) | UAS-Gbp2-RNAi | Vienna *Drosophila* resource center | VDRC: GD16696 | |
| Genetic reagent (*Drosophila melanogaster*) | UAS-CCHa2-RNAi | Vienna *Drosophila* resource center | VDRC: KK102257 | |
| Genetic reagent (*Drosophila melanogaster*) | UAS-mth-RNAi | Vienna *Drosophila* resource center | VDRC: KK106399 | |
| Genetic reagent (*Drosophila melanogaster*) | dilp2 | *Grönke et al., 2010* | | |
| Genetic reagent (*Drosophila melanogaster*) | $pten^{2L100}$ | *Oldham et al., 2002* | | |
| Genetic reagent (*Drosophila melanogaster*) | UAS-sun | *Delanoue et al., 2016* | | |
| Genetic reagent (*Drosophila melanogaster*) | $tra^{KO}$ | *Hudry et al., 2016* | | |
| Genetic reagent (*Drosophila melanogaster*) | $tra^{F\ K-IN}$ | *Hudry et al., 2019* | | |
| Genetic reagent (*Drosophila melanogaster*) | y,w;;ilp2HF | *Park et al., 2014* | | |
| Genetic reagent (*Drosophila melanogaster*) | tGPH (GFP-PH) | *Britton et al., 2002* | | |
| Genetic reagent (*Drosophila melanogaster*) | da-GAL4 | Bloomington *Drosophila* stock center | BDSC: 55849 | Note. Discontinued stock, equivalent stocks available |
| Genetic reagent (*Drosophila melanogaster*) | r4-GAL4 | Bloomington *Drosophila* stock center | BDSC: 33832 | |
| Genetic reagent (*Drosophila melanogaster*) | cg-GAL4 | Bloomington *Drosophila* stock center | BDSC: 7011 | |
| Genetic reagent (*Drosophila melanogaster*) | elav-GAL4 | Bloomington *Drosophila* stock center | BDSC: 458 | |
| Genetic reagent (*Drosophila melanogaster*) | dilp2-GAL4 | *Rulifson et al., 2002* | | |
| Antibody | Anti-sun guinea pig polyclonal | *Delanoue et al., 2016* | | (1:50) |
| Antibody | Anti-Cv-d guinea pig polyclonal | *Palm et al., 2012* | | (1:1000) |

*Continued on next page*

*Continued*

| Reagent type (species) or resource | Designation | Source or reference | Identifiers | Additional information |
|---|---|---|---|---|
| Antibody | Anti-pS6k rabbit polyclonal | Cell Signaling: 9209 | | (1:1000) |
| Antibody | Anti-Actin mouse monoclonal | Santa Cruz: 8432 | | (1:1000) |

### Fly husbandry

Larvae were raised at a density of 50 animals per 10 ml food at 25°C on *Drosophila* growth medium consisting of: 0.5x: 5.125 g/L sucrose, 17.725 g/L D-glucose, 12.125 g/L cornmeal, 11.325 g/L yeast, 4.55 g/L agar, 0.5 g CaCl$_2$•2H$_2$O, 0.5 g MgSO$_4$•7H$_2$O, 11.77 mL acid mix (propionic acid/phosphoric acid). 1x: 10.25 g/L sucrose, 25.45 g/L D-glucose, 24.25 g/L cornmeal, 22.65 g/L yeast, 4.55 g/L agar, 0.5 g CaCl$_2$•2H$_2$O, 0.5 g MgSO$_4$•7H$_2$O, 11.77 mL acid mix (propionic acid/phosphoric acid). 2x: 20.5 g/L sucrose, 70.9 g/L D-glucose, 48.5 g/L cornmeal, 45.3 g/L yeast, 4.55 g/L agar, 0.5 g CaCl$_2$•2H$_2$O, 0.5 g MgSO$_4$•7H$_2$O, 11.77 mL acid mix (propionic acid/phosphoric acid). 1Y: 20.5 g/L sucrose, 70.9 g/L D-glucose, 48.5 g/L cornmeal, 22.65 g/L yeast, 4.55 g/L agar, 0.5 g CaCl$_2$•2H$_2$O, 0.5 g MgSO$_4$•7H$_2$O, 11.77 mL acid mix (propionic acid/phosphoric acid). 2Y: 20.5 g/L sucrose, 70.9 g/L D-glucose, 48.5 g/L cornmeal, 45.3 g/L yeast, 4.55 g/L agar, 0.5 g CaCl$_2$•2H$_2$O, 0.5 g MgSO$_4$•7H$_2$O, 11.77 mL acid mix (propionic acid/phosphoric acid). Details for diets manipulating dietary sugar (1S) and calorie content (2Y calories) are found in *Supplementary file 3*. Our diets were also deposited in the *Drosophila* Dietary Composition Calculator (DDCC) (*Lesperance and Broderick, 2020*). Animals were collected as indicated in figure legends, and sexed by gonad size. When gonad size could not be used to determine sex (e.g. *tra* mutants, *da-GAL4>UAS-tra^F*), chromosomal females were identified by the presence of an X-linked GFP. Adult flies were maintained at a density of 20 flies per vial in single-sex groups.

### Fly strains

The following fly strains from the Bloomington *Drosophila* Stock Center were used: *Canton-S* (#64349), *w^1118* (#3605), *tra^1* (#675), *Df(3L)st-j7* (#5416), *srl^1* (#14965), *InR^E19* (#9646), TRiP control (#36303), *UAS-ilp2-RNAi* (#32475), *UAS-upd2-RNAi* (#33949), *UAS-tra^F* (#4590), *y,w* (#1495), *da-GAL4* (ubiquitous), *r4-GAL4* (fat body), *cg-GAL4* (fat body), *dilp2-GAL4* (IPCs), *elav-GAL4* (post-mitotic neurons), *UAS-rheb* (#9688), *UAS-cyt-c-p-RNAi* (#64898), *UAS-Idh-RNAi* (#41708), *mth^1* (#27896). The following fly strains from the Vienna *Drosophila* Resource Center were used in this study: *UAS-sun-RNAi* (GD23685), *UAS-Gbp1-RNAi* (KK108755) *UAS-Gbp2-RNAi* (GD16696), *UAS-CCHa2-RNAi* (KK102257), *UAS-mth-RNAi* (KK106399). Additional fly strains include: *dilp2* (*Grönke et al., 2010*), *pten^2L100*, *UAS-sun*, *tGPH* (GFP-PH), *tra^KO* (*Hudry et al., 2016*), *tra^F K-IN*(*Hudry et al., 2019*), *y,w;;ilp2HF* (*Park et al., 2014*). All genotypes used in the manuscript are listed in *Supplementary file 4*.

### Body size

Pupal volume was measured in male and female pupae as previously described (*Delanoue et al., 2010*; *Marshall et al., 2012*; *Rideout et al., 2012*; *Rideout et al., 2015*). For adult weight, 5-day-old virgin male and female flies were weighed in groups of 10 in 1.5 ml microcentrifuge tubes on an analytical balance. Wing length was measured as previously described (*Garelli et al., 2012*).

### Developmental timing

Larvae were placed into the experimental diet ±2 hr post-hatching. Percent pupation was calculated by comparing the number of pupae at 12 hr intervals to the total pupae in the vial after all animals pupated.

### Feeding behavior

Feeding behavior was quantified in sexed larvae by counting mouth hook contractions for 30 s.

## Protease feeding experiments

We treated larvae with a broad-spectrum protease inhibitor (PIC; Sigma-Aldrich #P2714) or a serine protease-specific inhibitor (AEBSF; Sigma-Aldrich #A8456) by adding the inhibitors to the food at final concentrations of 100 ml of 1x PIC per L, and 4 mM AEBSF as previously described (*Erkosar et al., 2015*).

## RNA extraction and cDNA synthesis

One biological replicate represents ten larvae frozen on dry ice and stored at −80℃. Each experiment contained three to four biological replicates per sex, per genotype, and per diet, and each experiment was repeated twice. RNA was extracted using Trizol (Thermo Fisher Scientific; 15596018) according to manufacturer's instructions, as previously described (*Marshall et al., 2012*; *Rideout et al., 2012*; *Rideout et al., 2015*; *Wat et al., 2020*). cDNA synthesis was performed using the QuantiTect Reverse Transcription Kit according to manufacturer's instructions (Qiagen; 205314).

## Quantitative real-time PCR (qPCR)

qPCR was performed as previously described (*Rideout et al., 2012*; *Rideout et al., 2015*; *Wat et al., 2020*). To determine changes in Foxo target gene expression, we plotted and analyzed the fold change in mRNA levels for each of three known Foxo target genes (*InR*, *bmm*, and *4E-BP*) together to quantify IIS activity in each sex and dietary context, an established approach to analyze co-regulated genes (*Blaschke et al., 2013*; *Hudry et al., 2019*). A complete primer list is available in *Supplementary file 5*.

## Preparation of protein extract

Dissected fat bodies were prepared for SDS-PAGE by homogenizing sets of ten larval fat bodies 108 hr after egg laying in an appropriate volume of lysis buffer (20 mM Hepes (pH 7.8), 450 mM NaCl, 25% glycerol, 50 mM NaF, 0.2 mM EDTA, 1 mM DTT, 1× Protease Inhibitor Cocktail (Roche, 04693124001), 1x Phosphatase Inhibitor Cocktail (Roche, 4906845001) using the Omni Bead Ruptor (VWR). Cellular fragments were pelleted, and supernatant collected by centrifugation for 5 min at 10,000 rpm at 4℃ (Thermo Scientific, Heraeus Pico 21 centrifuge). Protein concentration was determined by Bradford assay (Bio-Rad #550–0205) prior to SDS-PAGE.

## SDS-PAGE and Western blotting

A total of 20 μL of sample with 20 μg protein was loaded into each well. Proteins were separated using a 12% gel SDS-PAGE gel in SDS running buffer, and transferred to a nitrocellulose membrane (Bio-Rad) for 2 hr at 40 V on ice. Membranes were incubated for 1 hr in blocking buffer (5% milk or 5% BSA in TBST 0.1%) then incubated with primary antibodies overnight at 4℃. Membranes were washed (3 × 2 min) in TBST 0.1% then probed with secondary antibodies in blocking buffer for 1 hr at room temperature. After washes (3 × 2 min, 2 × 15 min, 1 × 5 min) in TBST 0.1%, membranes were treated with Pierce ECL (Thermo Scientific #32134) or Immobilon Forte (Millipore #WBLUF0100). Images were quantified using Image Studio (LI-COR). Primary antibodies: Anti-pS6K (#9209; Cell Signalling), and anti–actin (#8432; Santa Cruz), were used at 1:1000. HRP-conjugated secondary antibodies were used at 1:5000 for pS6k (anti-rabbit #65–6120; Invitrogen) and 1:3000 for actin (anti-mouse #7076; Cell Signalling).

## Hemolymph Western blotting

Hemolymph Western blotting was performed as previously described (*Delanoue et al., 2016*). Briefly, hemolymph from 40 larvae was collected in 40 μL of PBS with protease and phosphatase inhibitors (Roche 04693124001, Roche 4906845001), and hemocytes were removed by centrifugation according to the published protocol (*Delanoue et al., 2016*). Antibody concentrations used to detect hemolymph proteins were 1:50 for anti-Sun and 1:1000 for anti-Cv-d. Anti-guinea pig HRP-conjugated secondary was used at 1:2000.

## Fecundity and fertility

For female fecundity, single 6-day-old virgin female flies raised as indicated were crossed to three age-matched *CS* virgin males for a 24 hr mating period. Flies were transferred to fresh food vials

with blue 2Y food to lay eggs. The number of eggs laid over 24 hr was quantified. For male fertility, single 6-day-old virgin males were paired with three 6-day-old virgin *CS* females to mate, and females were allowed to lay eggs for 24 hr. The number of progeny was quantified by counting viable pupae.

## Microscopy

GFP-PH larvae were picked into 1Y or 2Y food. Larvae were dissected 108 hr after egg laying (AEL) and inverted carcasses were fixed for 30 min in 4% paraformaldehyde in phosphate buffered saline (PBS) at room temperature. Carcasses were rinsed twice with PBS, once in 0.1% Triton-X in PBS (PBST) for 5 min, then incubated with Hoechst (5 µg/mL, Life Technologies H3570), , and phalloidin fluor 647 (1:1000, Abcam ab176759) in PBST for 40 min. The stained carcasses were washed with PBS and mounted in SlowFade Diamond (Thermo Fisher Scientific S36972). Images were acquired with a Leica SP5 (20X). Mean GFP intensity was quantified at the cell membrane (marked by phalloidin) and in the cytoplasm using Fiji (*Schindelin et al., 2012*). Three cells per fat body were measured, and at least five fat bodies per sex and per diet were measured.

## Statistics and data presentation

Statistical analyses and data presentation were carried out using Prism GraphPad 6 (GraphPad Prism version 8.4.3 for Mac OS X). Statistical tests are indicated in figure legends and all *p*-values are listed in *Supplementary file 1*.

## Acknowledgements

We thank Dr. William Ja for the *UAS-sun* strain (*Delanoue et al., 2016*), Dr. Pierre Léopold for the anti-Sun antibody (*Delanoue et al., 2016*), Dr. Suzanne Eaton and Dr. Natalie Dye for the anti-Cv-d antibody (*Palm et al., 2012*), Dr. Bruce Edgar for the GFP-PH reporter (tGPH), and Dr. Linda Partridge for sharing the *dilp2* mutant strain. Stocks obtained from the Bloomington *Drosophila* Stock Center (NIH P40OD018537) were used in this study. We thank the TRiP at Harvard Medical School (NIH/NIGMS R01-GM084947) for providing transgenic RNAi fly stocks and/or plasmid vectors used in this study. Transgenic fly stocks and/or plasmids were also obtained from the Vienna *Drosophila* Resource Center (VDRC, http://www.vdrc.at). We acknowledge critical resources and information provided by FlyBase (*Thurmond et al., 2019*) FlyBase is supported by a grant from the National Human Genome Research Institute at the U.S. National Institutes of Health (U41 HG000739) and by the British Medical Research Council (MR/N030117/1). Funding for this study was provided by grants to EJR from the Canadian Institutes for Health Research (PJT-153072), Natural Sciences and Engineering Research Council of Canada (NSERC, RGPIN-2016–04249), Michael Smith Foundation for Health Research (16876), and the Canadian Foundation for Innovation (JELF-34879), and to IMA from the European Research Council (ERCAdG787470) and MRC Intramural Funding. JWM was supported by a 4 year CELL Fellowship from UBC, LWW was supported by a British Columbia Graduate Scholarship Award, ZS was supported by an NSERC Undergraduate Student Research Award, and BH was supported by an European Molecular Biology Organization Fellowship (aALTF782-2015). We would like to acknowledge that our research takes place on the traditional, ancestral, and unceded territory of the Musqueam people; a privilege for which we are grateful.

## Additional information

### Funding

| Funder | Grant reference number | Author |
|---|---|---|
| Canadian Institutes of Health Research | PJT-153072 | Elizabeth J Rideout |
| Natural Sciences and Engineering Research Council of Canada | RGPIN-2016-04249 | Elizabeth J Rideout |
| Michael Smith Foundation for Health Research | 16876 | Elizabeth J Rideout |

| Canada Foundation for Innovation | JELF-34879 | Elizabeth J Rideout |
| H2020 European Research Council | ERCAdG787470 | Irene Miguel-Aliaga |
| European Molecular Biology Organization | aALTF782-2015 | Bruno Hudry |
| University of British Columbia | CELL Fellowship | Jason W Millington |
| University of British Columbia | British Columbia Graduate Scholarship Award | Lianna W Wat |
| NSERC | Undergraduate Student Research Award | Ziwei Sun |
| MRC Intramural funding | | Irene Miguel-Aliaga |

The funders had no role in study design, data collection and interpretation, or the decision to submit the work for publication.

## Author contributions

Jason W Millington, Conceptualization, Formal analysis, Investigation, Visualization, Writing - original draft, Writing - review and editing; George P Brownrigg, Charlotte Chao, Formal analysis, Investigation; Ziwei Sun, Paige J Basner-Collins, Lianna W Wat, Investigation; Bruno Hudry, Irene Miguel-Aliaga, Resources; Elizabeth J Rideout, Conceptualization, Supervision, Funding acquisition, Validation, Writing - original draft, Project administration, Writing - review and editing

## Author ORCIDs

Jason W Millington  https://orcid.org/0000-0003-4330-2431
Elizabeth J Rideout  https://orcid.org/0000-0003-0012-2828

## Decision letter and Author response

Decision letter https://doi.org/10.7554/eLife.58341.sa1
Author response https://doi.org/10.7554/eLife.58341.sa2

# Additional files

## Supplementary files

- Supplementary file 1. Complete list of *p*-values for all experiments in this study.
- Supplementary file 2. Raw data values for all experiments in this study.
- Supplementary file 3. Details of 1S and 2Y calorie diets used in this study.
- Supplementary file 4. Complete list of *Drosophila melanogaster* genotypes used in this study.
- Supplementary file 5. Complete list of primers used in this study.
- Transparent reporting form

## Data availability

All data generated in this study are provided in Supplementary file 2. All statistical tests and p-values are listed in Supplementary file 1. Exact diets used in this study are described in Supplementary file 3 for ease of replication. All genotypes used in this study are listed in Supplementary file 4. A complete list of primers used in this study is provided in Supplementary file 5.

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
