## [Decision Letter]

**Acceptance summary:**

The insulin pathway differentially affects sex-specific phenotypes in most species, yet the molecular mechanisms underlying male-female differences in its regulation remain unclear. In this study, Rideout and colleagues investigated how females grow bigger in a protein-rich diet and unraveled that the sex determination gene, transformer, promotes a diet-induced trigger in Insulin signaling via stunted and srl/PGC-1a. This finding provides interesting genetic evidence for sex differences in complex phenotypes of development, physiology, and diseases.

**Decision letter after peer review:**

Thank you for submitting your article "Female-specific upregulation of insulin pathway activity mediates the sex difference in *Drosophila* body size plasticity" for consideration by *eLife*. Your article has been reviewed by three peer reviewers, one of whom is a member of our Board of Reviewing Editors, and the evaluation has been overseen by Utpal Banerjee as the Senior Editor. The reviewers have opted to remain anonymous.

The reviewers have discussed the reviews with one another and the Reviewing Editor has drafted this decision to help you prepare a revised submission.

Summary:

This is an interesting study addressing how sex-specific differences in endocrine signaling influences nutrient-dependent body size plasticity. The authors demonstrated that females, but not males, increase growth when raised on a high protein diet and revealed that growth difference results for sex-specific regulation of the insulin signaling pathway. Further, the authors identified that stunted expression is required for nutrient-dependent body size plasticity. Finally, the authors showed that the sex determination gene transformer is required for the female-specific expression of stunted on the high protein diet.

All the reviewers found the study very interesting, well designed and performed, and clearly written. Reviewers also agreed that the main finding of this study is novel and will be of broad interest in the field. However, at the same time, the reviewers raised several concerns about relatively weak points that need to be adequately addressed.

Essential revisions:

1) Identify the link between Tra and sun (related to reviewer 1 concerns 1-4, reviewer 2 concerns 4-5).

It will be critical to show the levels of circulating sun in females 1Y/2Y and males 1Y/2Y and/or the level of TOR signaling in the fat body in the four conditions.

2) Verify spargel mutant phenotype (e.g. sun expression in the spargel mutant) (related to reviewer 3).

3) Explain/repeat variable expressions of the IIS markers in males and females (reviewer 2 concerns 1-2).

4) Explain the relevance of 4EBP expression in r4>sun RNAi (2Y) condition (reviewer 2 concern 3).

5) Statistical analysis.

Some of these points may require additional experimental data and analyses. I have attached the original comments to clearly deliver reviewers' specific points.

Reviewer #1:

In this study, Rideout and colleagues investigated novel mechanisms underlying the female-specific size plasticity upon high protein diet and identified dilp2, sun, and tra as key molecules for the size control. Overall, experiments are well-performed, analyzed, and presented clearly. Also, the manuscript is well-written.

In the senior author's previous study, the author already showed a novel role for tra in the female body size determination through dilp2 in the IPCs under normal diet conditions. The function sun in the female-size plasticity is first shown and is the novel point of this work, and therefore, mechanisms involving tra-sun and sun-dilp2 require additional verifications. The manuscript describes the functions of tra, sun, or dilp2 on its own without showing the genetic/biochemical relationships of the three.

1) Is sun a direct transcriptional target of tra? The key question that needs to be addressed is whether tra functions upstream of sun and tra plays a role in the sun transcription.

2) Related to the above question, is sun the only target of tra? The authors investigated possible roles of humoral factors; however, downstream targets of tra may not be necessarily humoral factors.

3) No proof for the genetic relationship between tra-sun and sun-dilp2 is shown. For example, would overexpression of sun in tra mutant females rescue the nutrient-dependent growth?

4) Is mthl in the IPCs involved in this pathway?

5) To separate the body size difference and body size plasticity, it would be better to show the weight or volume changed (2Y minus 1Y) in different conditions for the plasticity, and absolute weight/volume numbers for growth.

6) It is interesting that InR/sun also controls egg production. However, without providing a detailed mechanism underlying this phenotype, this part makes the paper more complicated. Is tra also involved? Is this phenotype solely due to dilp2 and subsequent activation of InR in the ovary?

Reviewer #2:

This is an interesting study addressing the molecular basis of nutrient-dependent body size plasticity in *Drosophila*. The authors re-evaluate the sex-difference in nutritional plasticity in *Drosophila* and show that yeast (amino acids?) is the main component that drives plasticity in females, while males remain insensitive to increased yeast content (at least within the range of the experimental conditions used in the present study). They further imply insulin/IGF signaling (IIS) in this control, which is somehow expected. The novelty comes from the elucidation of the role of Stunted (Sun), a fat body factor controlling the level of circulating *Drosophila* insulin-like peptides (dilps), in the phenotypic plasticity observed in females. Whereas sun expression is increased in females raised on rich versus poor medium, this is not true in males. By knocking down sun in the fat body of larvae, they demonstrate the need for sun in this sex-specific regulation, and link it to the function of the sex-determination factor Transformer (Tra).

The data globally fits with the conclusions and the paper is rather convincing. It definitely brings a novel molecular twist to the interesting question of sex-specific nutritional plasticity. However, there are several issues with the experimental aspects that need to be corrected before the paper is ready.

1) Figure 2A,B: the markers for IIS show no variation in males fed 1Y or 2Y. However, this is not the case in the same experiments presented in Figure 4—figure supplement 2A, where Inr and 4E-BP go down in 2Y condition, as in females (see r4/+ and +>sun-RNAi controls). This casts doubts on the reproducibility of such analysis.

2) Figure 4—figure supplement 1C,D: statistical significance should compare +>sun-RNAi and r4>sun-RNAi, since they correspond to lower control values.

3) What is the significance of 4E-BP levels being increased in 2Y in r4>sun-RNAi conditions (both in females and males)? This would mean that IIS is generally reduced, which does not make sense. Therefore, what is the value of measuring 4E-BP as a marker for IIS?

4) Sun is secreted in the hemolymph and its circulating levels are controlled by TORC1 activity in FB cells (Delanoue et al., 2016). Therefore, an evaluation of circulating levels of Sun should be provided to better characterize female and male physiological responses to 1Y vs 2Y. Indeed, the results presented in Figure 4—figure supplement 4AB suggest that dysregulation of sun at the transcriptional level does not alter nutritional response in females and males. Looking at this figure, it is questionable whether nutritional plasticity is different in males and females of the r4>sun genotype.

5) Again, concerning Sun regulation, what is the link between Tra activiy and sun expression in response to Y content in females? Can the authors relay the level of Sun in 1Y vs 2Y to a difference in TOR activity specifically in female FB cells?

Reviewer #3:

This very well written manuscript by Millington et al., uses the fruit fly *Drosophila melanogaster* to understand how sex-specific differences in endocrine signaling influences nutrient-dependent body size plasticity. Through a series of well-designed experiments, the authors demonstrate that females, but not males, exhibit increase growth when raised on a diet with twice the nutrient diet of standard fly food. Through a series of logical experiments, the authors reveal that growth difference results for sex-specific regulation of the insulin signaling pathway – female, but not male, flies express increased levels of the humeral factor stunted, which is known to promote dilp2 secretion from the IPCs. The authors further demonstrate that the manner by which female flies regulated stunted expression is required for nutrient-dependent body size plasticity. Finally, the authors demonstrate that the sex determination gene transformer is required for female-specific expression of stunted on the high nutrient diet.

Overall, I found this a very nice story that works its way from a simple observation to a molecular mechanism. The story will be of broad interest and highlights the importance of studying sex-specific differences in animal growth and development. I have a few suggested revisions, but overall enjoyed reading the manuscript.

1) The authors used animals that are heterozygous for mutations in the gene spargel as a substitute for analyzing stunted mutants. While I understand the necessity of this experiment, spargel mutants have a wide range of metabolic defects that are independent of stunted and I'm concerned that this experiment requires a leap of faith. At a minimum, I'd like to see verification that heterozygous spargel mutants exhibit significant changes in stunted gene expression.

---

## [Author Response]

Essential revisions:1) Identify the link between Tra and sun (related to reviewer 1 concerns 1-4, reviewer 2 concerns 4-5).

We were also interested in the link between Tra and *sun*. To address Reviewer concerns, we completed several experiments, which we summarize below.

a) What is the link between Tra and *sun* mRNA levels?

We added a significant amount of data to the revised manuscript to indicate that *spargel* (*srl*), the *Drosophila* homolog of PGC-1a, represents one key link between Tra and the nutrient-dependent regulation of *sun* mRNA levels. In females, we show that heterozygous loss of *srl* blocks the nutrient-dependent upregulation of *sun* mRNA levels in a protein-rich diet (Figure 5A). In Tra-expressing males, heterozygous loss of *srl* similarly blocks the nutrient-dependent increase in *sun* mRNA levels (Figure 5D). Together, these experiments indicate that Srl is one link between Tra and the nutrient-dependent regulation of *sun* mRNA levels. Of note, *srl^1^* heterozygotes have no generalized growth defects (Figure 5B, C), suggesting these larvae do not have wide-ranging metabolic defects.

b) What is the genetic relationship between *tra* and *sun*?

In the revised version of our manuscript, we performed genetic epistasis experiments that suggest both Srl and *sun* lie downstream of Tra in regulating the sex difference in nutrient-dependent body size plasticity. For example, heterozygous loss of *srl* blocked the Tra-dependent upregulation of IIS activity in 2Y (Figure 5E) and eliminated the increased nutrient-dependent body size plasticity we observed in Tra-expressing males (Figure 5F).

Fat body-specific expression of a *UAS*-*sun-RNAi* transgene in Tra-expressing males similarly abolished the nutrient-dependent increase in body size (Figure 4H). Importantly, we rescued the smaller body size of *tra* mutant females simply by overexpressing *sun* in the fat body (Figure 4—figure supplement 2A). Together, these experiments support a model in which Tra influences phenotypic plasticity via Srl-mediated regulation of *sun* mRNA levels, which we communicate to readers by including a new figure summarizing our findings (Figure 7).

c) Is *sun* a direct Tra target?

Our discovery of a link between sex determination gene Tra and metabolic regulator Srl significantly advances our understanding of how Tra influences gene expression, as known Tra targets *dsx* and *fru* do not affect body size or several other Tra-regulated phenotypes (Rideout et al., 2015; Hudry et al., 2016; Garner et al., 2018). While we were not able to determine whether *sun* is a direct target of splicing factor Tra, we added text to the revised manuscript to highlight the importance of this question for future studies on Tra, Srl, and *sun*.

Importantly, while Srl targets other than *sun* were also regulated in a sex-specific and Tra-dependent manner (Figure 5—figure supplement 2A-D), two other Srl targets did not reproduce the effects of *sun* on phenotypic plasticity (Figure 5—figure supplement 2E-H). This strengthens our finding that *sun* is a key factor that impacts the sex difference in nutrient-dependent body size plasticity. However, to acknowledge that we cannot rule out all Srl targets in regulating phenotypic plasticity we added text to this effect in the revised manuscript (Discussion).

d) What is the role of *sun* mRNA levels in regulating body size?

Identifying Srl as the link between Tra and the regulation of *sun* mRNA levels suggests that an increase in *sun* mRNA levels should be able to increase body size. In our 1Y and 2Y diets, we found that *sun* overexpression in the fat body was sufficient to enhance body size in both sexes (Figure 3—figure supplement 7A, B). This finding indicates that increased *sun* mRNA levels are able to increase body size, and supports a model in which the nutrient-dependent increase in *sun* mRNA levels promotes growth to augment body size.

While this finding differs from results reported in a previous study (Delanoue et al., 2016), when we repeated their experiment using identical dietary conditions we found a male-specific increase in body size (Figure 3—figure supplement 8A). Interestingly, this increase in male body size was lost when we combined body size data from both sexes (Figure 3—figure supplement 8B). This indicates that the previous study failed to see a body size increase with *sun* overexpression due to a combination of dietary factors (we do not supplement food with live yeast to maintain tight control over nutrient content) and not analyzing body size data by sex. To clarify this point for readers, we added text to the revised manuscript (subsection “Increased nutrient-dependent body size plasticity in females promotes fecundity in a protein-rich context”).

e) Circulating Sun levels in males and females.

We were fortunate to receive enough anti-Sun antibody to perform one Western blot to measure hemolymph Sun levels in males and females (Figure 3—figure supplement 1A). Our quantification of the hemolymph Sun blot suggests that levels are nearly twice as high in females as they are in males reared in 2Y (Figure 3—figure supplement 1A). This finding supports a model in which higher levels of circulating Sun in females contribute to their larger body size.

f) Fat body TOR levels.

A previous study showed an important role for the Target-of-Rapamycin (TOR) pathway in mediating the nutrient-dependent secretion of Sun (Delanoue et al., 2016). When we measured fat body TOR activity using an antibody directed against a phosphorylated form of ribosomal protein S6 kinase (pS6k), a known TOR target, we saw no differences in pS6k levels between males and females in either the 1Y or the 2Y diet (Figure 5—figure supplement 1A-D). Similarly, when we examined changes to pS6k levels between control females and *tra* mutant females, there was no significant difference in fat body pS6k levels between females with and without Tra function on either diet (Figure 5—figure supplement 1F, G).

Therefore, while TOR activity undoubtedly plays a role in regulating Sun secretion, we did not find evidence of sex-specific or *tra*-dependent regulation of fat body TOR activity. This aligns with our previous finding that treating larvae with TOR inhibitor rapamycin does not have sex-biased effects on body size (Rideout et al., 2015). Because we show that changes to *sun* mRNA levels were sufficient to augment body size (Figure 3—figure supplement 7A, B), and that fat body TOR activity does not affect *sun* mRNA levels (Figure 5—figure supplement 1E), the data in our revised manuscript supports a role for the sex-specific regulation of *sun* mRNA levels via Srl as one mechanism underlying the male-female difference in body size plasticity.

2) Verify spargel mutant phenotype (e.g. sun expression in the spargel mutant) (related to reviewer 3).

Reviewer 3 raised important points about our use of *srl^1^*/+ larvae that we addressed in our revised manuscript. First, to ensure that heterogyzous loss of *srl* does not cause generalized developmental defects, we measured body size in 1Y. We found no body size reduction in *srl^1^*/+ larvae of either sex compared with *w1118* control larvae (Figure 5B, C). Given that body size encompasses the activity of many metabolic genes and pathways (Boulan et al., 2015), this suggests that *srl^1^*/+ larvae do not have generalized metabolic defects.

Second, we verified that heterozygous loss of *srl* blocks the nutrient-dependent upregulation of *sun* mRNA levels. In *w1118* females, we normally observe a nutrient-dependent upregulation of *sun* RNA in larvae raised on 2Y compared with larvae cultured on 1Y (Figure 5A); however, the nutrient-dependent increase in *sun* mRNA was blunted in *srl^1^*/+ females reared on 2Y compared with genotype-matched controls raised on 1Y (Figure 5A). Similarly, in Tra-expressing males (*da>tra^F^*), the nutrient-dependent increase in *sun* mRNA levels was blocked when those males were heterozygous for *srl^1^* (Figure 4F and Figure 5D).

Taken together, the data we present in our revised manuscript suggests that Srl plays a key role in the nutrient-dependent upregulation of *sun* mRNA levels in a protein-rich context, and that Srl mediates Tra’s effects on *sun* mRNA regulation.

3) Explain/repeat variable expressions of the IIS markers in males and females (reviewer 2 concerns 1-2).

We also noticed variability in levels of genes we used to quantify IIS activity between different male groups. In our original manuscript, we did not apply statistical tests to detect potential sex:diet and genotype:diet interactions in our gene expression data. In our revised manuscript, we improved our statistical analysis by applying these more rigorous tests to all of our gene expression data. Further, we identified an established way of displaying and analyzing co-regulated genes (*e.g*., Blaschke et al., 2013; Hudry et al., 2019) so that we no longer need to divide our panels displaying mRNA levels of Foxo target genes between the main and supplemental figures.

In our revised manuscript, we show that levels of Foxo target genes were significantly lower in all control females reared in 2Y compared with females reared in 1Y (Figure 1E, Figure 2A, Figure 3E, Figure 4A). Given that high levels of IIS activity repress Foxo target genes, this indicates higher IIS activity in females reared in 2Y. This improved statistical analysis also shows that the magnitude of the increase in IIS activity in *dilp2* mutant females, females with fat body loss of *sun*, and *tra* mutant females was significantly smaller than in control females (Figure 2A, Figure 3E, Figure 4A). This indicates that females with reduced *dilp2*, fat body *sun*, and *tra* were not able to upregulate IIS activity as much as control females.

When we repeated several key experiments in males, we still observed some variation in IIS readouts: males had either no change in Foxo target gene expression (Figure 1G), or a small but significant decrease in Foxo target genes (Figure 2B, Figure 3F). This suggested to us that males normally have a small but significant nutrient-dependent increase in IIS activity. To acknowledge this fact, we changed all instances of “female-specific” to “female-biased” in our revised manuscript when we refer to IIS activity.

Despite these minor differences among male genotypes, however, the most important conclusion we reached in our revised manuscript was that the magnitude of any nutrient-dependent changes to Foxo target genes in control males was always smaller than in genotype-matched females (sex:diet interactions in Supplementary file 1). This consistent and reproducible female-biased upregulation of IIS activity across all genotypes therefore supports one main finding of our paper: that a sex difference exists in the nutrient-dependent upregulation of IIS activity.

To display the female-biased decrease in Foxo target gene expression in a protein-rich diet more clearly, we included the % change in Foxo target gene expression for each genotype. This makes it is easier for readers to appreciate the sex difference in magnitude of nutrient-dependent changes to Foxo target gene expression.

4) Explain the relevance of 4EBP expression in r4>sun RNAi (2Y) condition (reviewer 2 concern 3).

*4E-BP* was one Foxo target gene among three Foxo targets that we measured to quantify IIS activity. In our revised manuscript, we sought a more rigorous way of analyzing gene expression for all three Foxo target genes so that we would not have to draw conclusions based on individual genes (e.g. 4E-BP). One established way we found to analyze co-regulated genes was to examine the behaviour of the genes as a group (Blaschke et al., 2013; Hudry et al., 2019). This way of analyzing gene expression allowed us to display Foxo target genes in a single graph, and to perform better statistical tests to detect genotype:diet and sex:diet interactions.

Using this improved statistical analysis, we showed that in *r4>+* and *+>UAS-sun-RNAi* females there was a significant decrease in Foxo target gene expression between 1Y and 2Y that was absent in *r4>UAS-sun-RNAi* females (Figure 3E). Given that there was a significant diet:genotype interaction (*p* < 0.0001), this suggests that the magnitude of change in Foxo target gene expression was different between *r4>UAS-sun-RNAi* females and *r4>+* and *+>UAS-sun-RNAi* controls. In contrast, there was no significant diet:genotype interaction between *r4>UAS-sun-RNAi* males and control males (*p* = 0.1068), indicating that Foxo target gene expression was not different in *r4>sun-RNAi* males compared with *r4>+* and *+>UAS-sun-RNAi* control males (Figure 3F).

When we examined sex:diet interactions, we found that the magnitude of the nutrient-dependent change to Foxo target genes was greater in females than males for the *r4>+* and *+>UAS-sun-RNAi* genotype, but not the *r4>UAS-sun-RNAi* genotype (*p* = 0.0166, 0.0119, and 0.1121, respectively). Thus, our improved statistical analysis of gene expression data indicates that the loss of fat body *sun* blocks the nutrient-dependent increase in IIS activity in females, but not in males.

5) Statistical analysis.

We thank the reviewer for pointing out that our original figures did not clearly communicate the fact that differences were only indicated as significant if the experimental genotype (*e.g*. *r4>sun-RNAi*) was significantly different from all control genotypes (*e.g*. *r4>+* and *+>sun-RNAi*). In our revised manuscript, we added lines to each graph to show all statistical comparisons that were made. The *p*-values for all multiple comparisons can be found in Supplementary file 1.

Reviewer #1:In this study, Rideout and colleagues investigated novel mechanisms underlying the female-specific size plasticity upon high protein diet and identified dilp2, sun, and tra as key molecules for the size control. Overall, experiments are well-performed, analyzed, and presented clearly. Also, the manuscript is well-written.In the senior author's previous study, the author already showed a novel role for tra in the female body size determination through dilp2 in the IPCs under normal diet conditions. The function sun in the female-size plasticity is first shown and is the novel point of this work, and therefore, mechanisms involving tra-sun and sun-dilp2 require additional verifications. The manuscript describes the functions of tra, sun, or dilp2 on its own without showing the genetic/biochemical relationships of the three.

We thank the reviewer for their thoughtful comments on our manuscript, and for their suggestion that we examine the relationships between *tra*, *sun*, and *dilp2* in more detail. We describe specific experiments in detail below.

1) Is sun a direct transcriptional target of tra?

For the sake of clarity, we answered this question below our two-part comments on whether *tra* functions upstream of *sun* transcription.

The key question that needs to be addressed is whether tra functions upstream of sun

In our original paper, we showed that females lacking *tra* were unable to upregulate *sun* mRNA levels when raised in a protein-rich diet (Figure 4B), whereas Tra expression in males was sufficient to enable the nutrient-dependent increase in *sun* mRNA levels in the 2Y diet (Figure 4F). While this suggests that *tra* may lie upstream of *sun*, in our revised manuscript we used a genetic approach to strengthen our conclusions about the relationship between Tra and *sun*. Further, we identify Srl as one link between Tra and *sun* mRNA levels (next point).

In our revised manuscript, we show that fat body-specific *sun* knockdown blocked the nutrient-dependent increase in body size we observed in Tra-expressing males (Figure 4H). Further, we demonstrate that fat body *sun* overexpression was sufficient to restore the smaller body size of *tra* mutant females raised in a protein-rich diet (Figure 4—figure supplement 2A), suggesting that *sun* lies downstream of Tra in promoting body size in this context.

Together, these new data support a model in which Tra promotes increased body size in a protein-rich context by acting upstream of *sun*. To ensure that these findings are clearly communicated to the reader, we included these new data and a new summary figure to the revised manuscript (Figure 7).

and tra plays a role in the sun transcription.

We were also curious about the link between Tra and regulation of *sun* mRNA levels, as two transcription factors known to mediate Tra’s gene expression effects do not affect body size (*doublesex [dsx]* and *fruitless* [*fru]*; Rideout et al., 2015). In our revised manuscript, we identify *spargel* (*srl*), the *Drosophila* homolog of PGC-1a, as one link between Tra and regulation of *sun* mRNA levels.

A previous study showed that Srl regulates *sun* mRNA levels in response to dietary protein (Delanoue et al., 2016), which we confirmed by showing that female larvae heterozygous for *srl^1^* were unable to augment *sun* mRNA levels in a protein-rich context (Figure 5A). This suggests that normal Srl function was required for the nutrient-dependent upregulation of *sun* mRNA in females. Importantly, we confirmed that the *srl^1^*/+ larvae show no obvious developmental defects (Figure 5B, C).

To determine whether Srl mediates Tra’s effects on *sun* mRNA levels in a protein-rich context, we monitored nutrient-dependent changes to *sun* mRNA levels in Tra-expressing males, and in Tra-expressing males carrying the *srl^1^* allele. While males with ectopic Tra expression normally upregulate *sun* mRNA levels in a protein-rich context (Figure 4F), heterozygous loss of Srl function in these Tra-expressing males blocked the nutrient-dependent upregulation of *sun* mRNA (Figure 5D). This suggests that Srl function mediates Tra’s effects on *sun* mRNA levels, identifying a new mechanism by which Tra affects gene expression.

Because Tra-expressing males carrying the *srl^1^* allele no longer increase IIS activity or body size in a protein-rich context (Figure 5E, F), in contrast to Tra-expressing males with normal Srl function (Figure 4E, G), the data in our revised manuscript suggests that the regulation of *sun* mRNA levels by Srl plays an important role in mediating Tra’s effects on phenotypic plasticity. To reflect the importance of this new data, we added these figures, a significant amount of new text, a new model figure (Figure 7), and a new section to the Discussion.

Is sun a direct transcriptional target of tra?

Much of our knowledge about how Tra regulates gene expression comes from studies on how Tra, a splicing regulator, controls the sex-specific splicing of pre-mRNA of transcription factors *dsx* and *fru*. Because Tra impacts body size independently of these two genes (Rideout et al., 2015), our discovery of transcriptional coactivator Srl as one link between Tra and regulation of *sun* mRNA levels reveals a new way in which splicing factor Tra impacts gene expression.

While we adjusted the text to reflect the fact that the precise biochemical interactions between Tra, Srl, and *sun* require further study, as Tra is a splicing factor and Srl is a transcriptional coactivator that partners with multiple transcription factors to influence transcript levels (Tiefenbock et al., 2010) (Discussion), we also added text to highlight how uncovering the Tra-Srl link advances our understanding of Tra-dependent changes to gene expression independently of *dsx* and *fru* (Discussion).

For example, there is a rapidly growing body of literature describing Tra-dependent but *dsx*- and *fru*-independent effects on lifespan, organ plasticity, and neural circuits (Rideout et al., 2015; Hudry et al., 2016; Regan et al., 2016; Castellanos et al., 2013; Garner et al., 2018); however, the molecular mechanisms underlying these *dsx*- and *fru*-independent effects remain unknown. Our findings will therefore help researchers studying sex differences in traits such as lifespan, organ plasticity, and neural circuits by providing new insight into the mechanisms underlying Tra-dependent but *dsx*- and *fru*-independent effects on gene expression.

2) Related to the above question, is sun the only target of tra? The authors investigated possible roles of humoral factors; however, downstream targets of tra may not be necessarily humoral factors.

We previously showed that *tra’s* only confirmed direct downstream targets *dsx* and *fru* do not affect body size (Rideout et al., 2015); however, given that Srl mediates the nutrient-dependent upregulation of *sun* mRNA levels downstream of Tra, we also looked at gene expression changes in other known Srl targets. Like *sun*, we found that other confirmed Srl targets were regulated in a sex-specific and nutrient-dependent manner (Figure 5—figure supplement 2A-D). Yet when we used RNAi to knock down levels of Srl targets other than *sun*, loss of two additional Srl target genes that are functionally similar to *sun* did not affect phenotypic plasticity (Figure 5—figure supplement 2E-H). Note: loss of fat body *bellwether* (*blw*; FBgn0011211) and *cytochrome c oxidase subunit 5a* (*Cox5a*; FBgn0019624) did not produce viable animals for our measurements.

While this data suggests that *sun* plays an important role among Srl targets in mediating sex-specific body size plasticity downstream of Tra, we adjusted the text in our revised manuscript to reflect the fact that we cannot rule out all possible Srl targets in regulating sex-specific or nutrient-dependent growth.

3) No proof for the genetic relationship between tra-sun and sun-dilp2 is shown. For example, would overexpression of sun in tra mutant females rescue the nutrient-dependent growth?

We include this important experiment in our revised manuscript, and confirm that the overexpression of fat body *sun* in a *tra* mutant female is able to fully rescue the smaller body size in these females when they are raised in a protein-rich diet (Figure 4—figure supplement 2A). Further, we show that loss of fat body *sun* in a Tra-expressing male blocks the increased phenotypic plasticity we normally observe in males with ectopic Tra expression (Figure 4H). Together, these results strengthen a model in which *sun* lies downstream of Tra in regulating nutrient-dependent changes to body size. To address the *sun*-Dilp2 relationship, in our revised manuscript we highlight the elegant and comprehensive work that was done in the Léopold lab to show that *sun* impacts body size via regulation of Dilp2 secretion to ensure that the reader is aware of this important body of work.

4) Is mthl in the IPCs involved in this pathway?

To address a role for *mth* in the insulin-producing cells (IPCs) in the brain in this pathway, we used *dilp2-GAL4* to knock down *mth* levels using RNAi. Unlike all other control strains we examined in this study, the *dilp2-GAL4>+* control strain did not show a sex difference in nutrient-dependent body size plasticity (sex:diet interaction, *p* = 0.9995). We therefore used two alternative approaches to investigate a potential role for *mth* function in phenotypic plasticity: RNAi-mediated knock down of *mth* in all neurons, and whole-body loss of *mth*. Neither pan-neuronal, nor global, loss of *mth* significantly affected phenotypic plasticity (Figure 3—figure supplement 5A-F). The apparent discrepancy with the previous study is most likely due to differences in lab diets, using different *dilp2-GAL4* lines, and the fact that we are not reproducing the original experiment with IPC-specific *mth* knockdown. While we included these new results in the revised manuscript (Figure 3—figure supplement 5A-F), we added text accompanying these figures to ensure the reader is aware of the limitations of our data compared with the original findings in Delanoue *et al.*, 2016.

5) To separate the body size difference and body size plasticity, it would be better to show the weight or volume changed (2Y minus 1Y) in different conditions for the plasticity, and absolute weight/volume numbers for growth.

We thank the reviewer for this suggestion. In order to ensure consistency between our data on pupal volume/adult weight data and past papers on Sun (Delanoue et al., 2016), we chose to maintain our current data presentation style. This will ensure that the audience is easily able to compare our data with past literature on Sun, and more generally within the larval growth field.

6) It is interesting that InR/sun also controls egg production. However, without providing a detailed mechanism underlying this phenotype, this part makes the paper more complicated. Is tra also involved? Is this phenotype solely due to dilp2 and subsequent activation of InR in the ovary?

We were also very interested in the link between nutrition, IIS and egg production. To ensure we acknowledge the body of literature in this area, we added text and citations to ensure that the reader is aware of past studies showing that increased circulating levels of Dilp proteins and nutrition can enhance egg production by increasing ovariole number. Further, we suggest future studies that should be completed in order to gain detailed insight into the fertility phenotypes that arise from changes to *InR*, *sun*, and *dilp2*.

Reviewer #2:This is an interesting study addressing the molecular basis of nutrient-dependent body size plasticity in Drosophila. The authors re-evaluate the sex-difference in nutritional plasticity in *Drosophila* and show that yeast (amino acids?) is the main component that drives plasticity in females, while males remain insensitive to increased yeast content (at least within the range of the experimental conditions used in the present study). They further imply insulin/IGF signaling (IIS) in this control, which is somehow expected. The novelty comes from the elucidation of the role of Stunted (Sun), a fat body factor controlling the level of circulating *Drosophila* insulin-like peptides (dilps), in the phenotypic plasticity observed in females. Whereas sun expression is increased in females raised on rich versus poor medium, this is not true in males. By knocking down sun in the fat body of larvae, they demonstrate the need for sun in this sex-specific regulation, and link it to the function of the sex-determination factor Transformer (Tra).The data globally fits with the conclusions and the paper is rather convincing. It definitely brings a novel molecular twist to the interesting question of sex-specific nutritional plasticity. However, there are several issues with the experimental aspects that need to be corrected before the paper is ready.

We thank the reviewer for their careful reading and analysis of the paper. We address specific points and suggestions for improvement below.

1) Figure 2A,B: the markers for IIS show no variation in males fed 1Y or 2Y. However, this is not the case in the same experiments presented in Figure 4—figure supplement 2A, where Inr and 4E-BP go down in 2Y condition, as in females (see r4/+ and +>sun-RNAi controls). This casts doubts on the reproducibility of such analysis.

In our original manuscript, we did not apply statistical tests to detect potential sex:diet and genotype:diet interactions in our gene expression data. In our revised manuscript, we improved our statistical analysis by applying these more rigorous tests to all of our gene expression data. Further, we identified an established way of displaying and analyzing co-regulated genes (*e.g*., Blaschke et al., 2013; Hudry et al., 2019) so that we no longer need to divide our panels displaying mRNA levels of Foxo target genes between the main and supplemental figures.

These changes allowed us to make more accurate conclusions about the behavior of Foxo target genes in response to variables such as sex, diet, and genotype. Also, we arranged the data in a better way to highlight the female-biased decrease in Foxo target gene expression in a protein-rich diet. Additional measures to help the reader appreciate the sex difference in the magnitude of gene expression changes include displaying the % change in Foxo target gene expression for each genotype.

In our revised manuscript we show that levels of Foxo target genes were significantly lower in all control females reared in 2Y compared with females reared in 1Y (Figure 1E, Figure 2A, Figure 3E, Figure 4A, Figure 4—figure supplement 3A). Given that high levels of IIS activity repress Foxo target genes, this indicates higher IIS activity in 2Y, as we previously showed. In males, when we repeated several key experiments, we still observed some variation in IIS readouts: we found either no change in Foxo target gene expression (Figure 1G), or a small but significant decrease in Foxo target genes (Figure 2B, Figure 3F). This suggests that males normally have a small but significant nutrient-dependent increase in IIS activity. To acknowledge this fact, we changed all instances of “female-specific” to “female-biased” in our revised manuscript when we refer to IIS activity.

Despite these minor differences among male genotypes, however, the most important conclusion we reached in our revised manuscript was that the magnitude of any nutrient-dependent change to Foxo target genes in control males was always smaller than in genotype-matched females (sex:diet interactions in Supplementary file 1). This reproducible female-biased upregulation of IIS activity across all genotypes therefore supports our finding that a sex difference exists in the nutrient-dependent upregulation of IIS activity.

With respect to the *r4>UAS-sun-RNAi* experiment, we show that *r4>+* and *+>sun-RNAi* males have a significant decrease in mRNA levels of Foxo target genes and *r4>sun-RNAi* males males do not; however, there was no significant genotype:diet interaction among *r4>+*, *+>sun-RNAi,* and *r4>sun-RNAi* males (*p* = 0.1068). This indicates that there was no effect of genotype on Foxo target gene expression in males (Figure 3F), in contrast to the significant genotype:diet interaction we observed in females (*p* < 0.0001) (Figure 3E). Importantly, the magnitude of the reduction in Foxo target gene expression in control males was smaller than in genotype-matched females (sex:diet interactions *p* = 0.0166 and 0.0119, respectively), but not different between *r4>sun-RNAi* males and females (*p* = 0.1121).

Overall, we obtained a more accurate picture of nutrient-dependent changes to IIS activity in each sex and genotype, across two diets, by applying more rigorous statistical tests to our analysis of Foxo target genes. Together, we believe these changes support our conclusion that there is a female-biased increase in IIS activity in response to dietary protein, and that this nutrient-dependent increase in IIS activity requires *dilp2*, *sun*, and *tra* function.

2) Figure 4—figure supplement 1C,D: statistical significance should compare +>sun-RNAi and r4>sun-RNAi, since they correspond to lower control values.

We thank the Reviewer for highlighting that our graphs do not clearly indicate that our statistical analyses make multiple comparisons between all three genotypes included in this, and other, experiments. We have adjusted all of the graphs in the manuscript to show the statistical comparisons between all the genotypes for the sake of clarity.

3) What is the significance of 4E-BP levels being increased in 2Y in r4>sun-RNAi conditions (both in females and males)? This would mean that IIS is generally reduced, which does not make sense. Therefore, what is the value of measuring 4E-BP as a marker for IIS?

We also wondered about why 4E-BP for some genotypes does not correspond with the behavior of other Foxo target genes. We therefore sought a way of analyzing the Foxo target genes together to gain a better picture of Foxo activity than we would obtain by drawing conclusions based on the behaviour of single genes. One established way to analyze co-regulated genes is to examine the behavior of the genes as a group (Blaschke et al., 2013; Hudry et al., 2019), and applying rigorous statistical tests to detect genotype:diet and sex:diet interactions.

Using this improved statistical analysis, we showed that in *r4>+* and *+>UAS-sun-RNAi* females there was a significant decrease in Foxo target gene expression between 1Y and 2Y that was absent in *r4>UAS-sun-RNAi* females (Figure 3E). Given that there was a significant diet:genotype interaction (*p* < 0.0001), this suggests that the magnitude of change to Foxo target gene expression was different between *r4>UAS-sun-RNAi* females and *r4>+* and *+>UAS-sun-RNAi* controls. In contrast, there was no significant diet:genotype interaction between *r4>UAS-sun-RNAi* males and control males (*p* = 0.1068), indicating that Foxo target gene expression was not different in *r4>sun-RNAi* males compared with *r4>+* and *+>UAS-sun-RNAi* control males (Figure 3F).

Importantly, sex:diet interactions showed that the magnitude of the nutrient-dependent change to Foxo target genes was greater in females than males for the *r4>+* and *+>UAS-sun-RNAi* genotype, but not the *r4>UAS-sun-RNAi* genotype (*p* = 0.0166, 0.0119, and 0.1121, respectively). Thus, our improved statistical analysis of gene expression data for this experiment supports our conclusion that the loss of fat body *sun* blocks the nutrient-dependent increase in IIS activity in females, but not in males.

4) Sun is secreted in the hemolymph and its circulating levels are controlled by TORC1 activity in FB cells (Delanoue et al., 2016). Therefore, an evaluation of circulating levels of Sun should be provided to better characterize female and male physiological responses to 1Y vs 2Y.

We were also curious about circulating Sun levels, and a potential role for TOR in males and females on both diets. We first measured hemolymph Sun levels in males and females raised on 1Y and 2Y. We were fortunate to receive enough antibody to make 0.5 ml of primary antibody solution to perform this Western blot. We found that hemolymph levels of Sun were approximately twice as high in males as in females (Figure 3—figure supplement 1A), suggesting that there is a sex difference in circulating Sun.

To determine whether fat body TOR may affect the sex difference in circulating Sun, we measured TOR activity in fat bodies isolated from males and females reared in 1Y and 2Y. We found that there were no sex differences in phospho-S6k (pS6k) levels, an established readout for TOR signaling, in either diet (Figure 5—figure supplement 1A-D). Similarly, we found no significant changes to pS6k levels between fat bodies isolated from control and *tra* mutant females (Figure 5—figure supplement 1F, G).

This suggests that TOR activity does not normally differ between males and females, or between control females and *tra* mutant females, indicating that TOR may not be the sole determinant of circulating Sun levels. Given that we show fat body *sun* overexpression increases body size in multiple nutritional contexts when larvae are analyzed according to sex (Figure 3—figure supplement 7A, B, Figure 3—figure supplement 8A), it will be interesting to test whether increased *sun* mRNA promotes circulating Sun levels. Anti-Sun antibody quantities are too limiting for us to do this experiment at present; however, we added text to highlight the importance of this experiment (subsection “A nutrient-dependent increase in stunted mRNA levels is required for enhanced IIS activity and a larger body size plasticity in females cultured in a protein-rich context”).

Indeed, the results presented in Figure 4—figure supplement 4AB suggest that dysregulation of sun at the transcriptional level does not alter nutritional response in females and males. Looking at this figure, it is questionable whether nutritional plasticity is different in males and females of the r4>sun genotype.

Given that our data suggests it is the ability to augment *sun* mRNA levels, rather than absolute *sun* mRNA levels, that forms the basis of phenotypic plasticity, we measured *sun* mRNA levels in 1Y and 2Y in male larvae with fat body *sun* overexpression. We found no nutrient-dependent change in *sun* expression in these males (Figure 3—figure supplement 8C). Thus, the likely reason that *sun* overexpression does not enhance phenotypic plasticity in males was that the nutrient-dependent increase in *sun* mRNA levels was still absent in this overexpression context. To more clearly communicate these key points to the readers we have added text to this effect in the revised manuscript (subsection “A nutrient-dependent increase in stunted mRNA levels is required for enhanced IIS activity and a larger body size plasticity in females cultured in a protein-rich context”), and included a graphical abstract summarizing our model (Figure 7).

5) Again, concerning Sun regulation, what is the link between Tra activiy and sun expression in response to Y content in females?

We thank the reviewer for the opportunity to identify the link between Tra and *sun* mRNA levels. We added a significant amount of data in the revised manuscript to indicate that *spargel* (*srl*), the *Drosophila* homolog of PGC-1a, represents one link between Tra and the diet-dependent regulation of *sun* mRNA levels.

A previous study showed Srl mediates the nutrient-dependent upregulation of *sun* mRNA levels in a mixed-sex population of larvae (Delanoue et al., 2016). In our revised manuscript, we confirmed Srl function is required for the nutrient-dependent increase in *sun* mRNA levels in females raised on 2Y: the nutrient-dependent upregulation of *sun* mRNA was blocked in females heterozygous for the strong hypomorphic *srl^1^* allele (Figure 5A).

Further, we show that heterozygous loss of *srl* blocks the nutrient-dependent upregulation of *sun* mRNA levels in Tra-expressing males (Figure 5D). Given that heterozygous loss of *srl* also blocks the nutrient-dependent upregulation of IIS activity (Figure 5E) and increased body size of Tra-expressing males raised in a protein-rich diet (Figure 5F), our data suggests that Srl represents one important link between Tra and nutrient-dependent changes to *sun* mRNA levels, IIS activity, and phenotypic plasticity. To ensure the reader is aware of this new data, we added several figures, text, and discussion (subsection “Transcriptional coactivator Spargel represents one link between Transformer and regulation of *sun* mRNA levels”) on this topic to the revised manuscript.

Can the authors relay the level of Sun in 1Y vs 2Y to a difference in TOR activity specifically in female FB cells?

TOR has been shown to play an important role in regulating Sun secretion, and we confirm in the revised version of our manuscript that there is a sex difference in circulating Sun (Figure 3—figure supplement 1A, B). To determine whether TOR plays a role in regulating Sun release, we measured fat body TOR activity by quantifying pS6k levels in males and females in 1Y and 2Y. We found that fat body pS6k levels did not differ between the sexes in 1Y or 2Y (Figure 5—figure supplement 1A-D). Similarly, we found no significant difference in pS6k levels between control and *tra* mutant females (Figure 5—figure supplement 1F, G).

Given that we detected no sex differences in fat body TOR activity, and that increased TOR does not alter *sun* mRNA levels (Figure 5—figure supplement 1E), our data supports a model in which the sex-specific and Tra-dependent regulation of *sun* occurs primarily through Srl rather than TOR. To clarify this model, we added data and text to the revised manuscript to ensure the reader understands our model of sex-specific body size plasticity (Figure 7; subsection “A nutrient-dependent increase in stunted mRNA levels is required for enhanced IIS activity and a larger body size plasticity in females cultured in a protein-rich context”). Further, we added text to discuss how this finding aligns with our previous finding that rapamycin feeding in larvae did not impact sexual size dimorphism (Rideout et al., 2015) (subsection “A nutrient-dependent increase in stunted mRNA levels is required for enhanced IIS activity and a larger body size plasticity in females cultured in a protein-rich context”).

Reviewer #3:This very well written manuscript by Millington et al., uses the fruit fly *Drosophila melanogaster* to understand how sex-specific differences in endocrine signaling influences nutrient-dependent body size plasticity. Through a series of well-designed experiments, the authors demonstrate that females, but not males, exhibit increase growth when raised on a diet with twice the nutrient diet of standard fly food. Through a series of logical experiments, the authors reveal that growth difference results for sex-specific regulation of the insulin signaling pathway – female, but not male, flies express increased levels of the humeral factor stunted, which is known to promote dilp2 secretion from the IPCs. The authors further demonstrate that the manner by which female flies regulated stunted expression is required for nutrient-dependent body size plasticity. Finally, the authors demonstrate that the sex determination gene transformer is required for female-specific expression of stunted on the high nutrient diet.Overall, I found this a very nice story that works its way from a simple observation to a molecular mechanism. The story will be of broad interest and highlights the importance of studying sex-specific differences in animal growth and development. I have a few suggested revisions, but overall enjoyed reading the manuscript.

We thank the reviewer for their positive assessment of our manuscript. We address each suggestion for improvement below.

1) The authors used animals that are heterozygous for mutations in the gene spargel as a substitute for analyzing stunted mutants. While I understand the necessity of this experiment, spargel mutants have a wide range of metabolic defects that are independent of stunted and I'm concerned that this experiment requires a leap of faith. At a minimum, I'd like to see verification that heterozygous spargel mutants exhibit significant changes in stunted gene expression.

Reviewer 3 makes very good points about verifying the effects of Srl on *sun* mRNA levels, and ensuring no broad metabolic defects exist in *srl* larvae. In our revised manuscript, we show that the nutrient-dependent increase in *sun* mRNA levels was blocked in female *srl^1^*/+ larvae raised a protein-rich diet (Figure 5A).

To determine whether heterozygous loss of Srl also impacts *sun* regulation in the context of Tra overexpression, we measured *sun* levels in Tra-expressing male larvae (*da-GAL4*>*UAS-tra*). Normally, these Tra-expressing males show nutrient-dependent upregulation of *sun* mRNA (Figure 4F); however, in our revised manuscript we show heterozygous loss of *srl* in these Tra-expressing males blocks their ability to augment *sun* mRNA in a protein-rich context (Figure 5D).

Importantly, the changes we observe are unlikely to be caused by generalized metabolic defects, as body size in *srl^1^*/+ larvae was not reduced compared with a control strain in the 1Y diet (Figure 5B, C). Together, the new data we present suggests that the ability of Tra to promote phenotypic plasticity depends on the regulation of *sun* mRNA levels by Srl, as loss of this regulation abolishes the ability of Tra-expressing males to upregulate *sun*, IIS activity, and body size in a protein-rich context (Figure 5D-F). Indeed, loss of *sun* in Tra-expressing males and females blocks phenotypic plasticity (Figure 4H, Figure 4—figure supplement 3H). To ensure readers are aware of all these important points, we include text and new data on this topic in the revised manuscript (Results).